

# Spatial and temporal CCN variations in convection-permitting aerosol microphysics simulations in an idealised marine tropical domain

Céline Planche[1,*], Graham W. Mann[1,2], Kenneth S. Carslaw[1], Mohit Dalvi[3], John H. Marsham[1,2], and Paul R. Field[1,3]

[1] School of Earth and Environment, University of Leeds, Leeds, United Kingdom
[2] National Centre for Atmospheric Science, United Kingdom
[3] Met Office, Exeter, United Kingdom
*Now at: Laboratoire de Météorologie Physique, Université Clermont Auvergne, OPGC/CNRS UMR 6016, Clermont-Ferrand, France

*Correspondence to:* Céline Planche (C.Planche@opgc.univ-bpclermont.fr) and Graham Mann (G.W.Mann@leeds.ac.uk)

**Abstract.** A convection-permitting limited area model with periodic lateral boundary conditions and prognostic aerosol microphysics is applied to investigate how concentrations of cloud condensation nuclei (CCN) in the marine boundary layer are affected by high resolution dynamical and thermodynamic fields. The high-resolution aerosol microphysics–dynamics model, which resolves differential particle growth and aerosol composition across the particle size range, is applied on a domain designed to match approximately a single grid square of a climate model. We find that, during strongly convective conditions, CCN concentrations vary by more than a factor of 8 across the domain (5th-95th percentile range), and a factor of ~3 at more moderate wind-speed conditions. One reason for these large sub-climate-grid-scale variations in CCN is that emissions of sea-salt and DMS are much higher when spatial and temporal wind speed fluctuations become resolved at this convection-permitting resolution (making peak wind speeds higher). By analysing how the model evolves during spin-up, we gain new insight into the way primary sea-salt and secondary sulphate particles contribute to the overall CCN variance in these realistic conditions, and find a marked difference in the variability of super-micron and sub-micron CCN. Whereas the super-micron CCN are highly variable, being dominated by strongly fluctuating emitted sea-spray, the sub-micron CCN tend to be steadier, being mainly produced on longer timescales following growth after new particle formation in the free troposphere, with fluctuations inherently buffered by the fact that coagulation is faster at higher particle concentrations. We also find that sub-micron CCN are less variable in particle size, the accumulation mode mean size varying by ~20% (0.101 to 0.123 μm diameter) compared to ~35% (0.75 to 1.10 μm diameter) for coarse mode particles at this resolution. We explore how the CCN variability changes in the vertical, and at different points in the spin-up, showing how CCN concentrations are introduced both by the emissions close to the surface, and at higher altitudes during strongly convective conditions. We also explore how the non-linear variation of sea-salt emissions with wind speed propagates into variations in sea-salt mass mixing ratio and CCN concentrations, finding less variation in the latter two quantities due to the longer



transport timescales inherent with finer CCN, which sediment more slowly. The complex mix of sources and diverse community of processes involved makes sub-grid parameterization of CCN variations difficult. However, the results presented here illustrate the limitations of predictions with large-scale models and the high-resolution aerosol-dynamics modelling system shows promise for future studies where the aerosol variations will propagate through to modified cloud
microphysical evolution.

**Keywords.** Aerosol particles, CCN variability, UKCA, GLOMAP-mode, sub-climate scale, idealised marine tropical case.

## 1 Introduction

Aerosol particles affect the Earth's climate system directly by scattering and absorbing short-wave and long-wave radiation and indirectly by influencing the albedo and lifetime of clouds (e.g. Lohmann and Feichter, 2005). Successive IPCC climate
assessment reports (e.g. Forster et al., 2007; Myhre et al., 2013) have identified the radiative forcing due to aerosol-climate interactions as having a high level of uncertainty that needs to be better constrained for improved prediction of anthropogenic climate change.

Atmospheric aerosols, whether natural or anthropogenic, originate from two different pathways: directly emitted "primary particles" (e.g. sea-spray, in marine environments) and secondary particles, which are formed by nucleation, often first
requiring oxidation of gaseous precursors such as dimethyl sulphide (DMS). In general, the primary particle population can be straightforwardly classified into natural (dust, sea-spray, primary biogenic) or anthropogenic (e.g. carbonaceous particles from fossil-fuel combustion sources). However, this classification is not possible for secondary particles because of the complex interactions and influences of gases with both natural and anthropogenic sources (such as sulphur dioxide) and the moderating influence of additional semi-volatile species such as ammonia and nitric acid. In the marine boundary layer
however, the dominant two sources of cloud condensation nuclei (CCN) are DMS and sea-spray (e.g. Raes et al., 1993; O'Dowd and de Leeuw, 2007; Boucher et al., 2013) and the relative simplicity of this particular compartment of the atmosphere allows the systematic assessment of how two types of natural particles: primary sea-spray and secondary sulphate particles from DMS, influence aerosol-cloud interactions. Carslaw et al. (2013) highlight the importance of quantifying such natural aerosols in order to accurately characterise the anthropogenic radiative forcing via aerosol-cloud
interactions.

Until recently, computational costs have tended to constrain most climate models participating in international climate assessment reports to treat aerosol-cloud interactions in a simplified way, with only the mass of several aerosol types transported. With this conventional approach, CCN (number) concentrations are derived from the transported masses based on an assumed size distribution for each type, often taken to be globally uniform (e.g. Jones et al., 2011). The need to
represent aerosol-cloud interactions more realistically has been a major motivation for the development of a new generation of composition-climate models with interactive aerosol microphysics. The models transport both particle number





concentrations and component masses (e.g. sulphate, black carbon) in multiple size classes (e.g. Mann et al., 2014), and allow to represent sources of primary and secondary CCN explicitly. For example, the UK's Earth System Model for CMIP6 (Coupled Model Intercomparison Project phase 6) includes the GLOMAP (Global Model of Aerosol Processes) aerosol microphysics module (Mann et al., 2010; 2012), which resolves differential particle growth and aerosol composition across

the particle size range including internal mixtures via the computationally efficient modal aerosol dynamics approach.

In order to understand how aerosols and clouds interact, it is important to assess how aerosol properties vary at finer spatial scales than are resolved in climate models, where both convective-dynamical and aerosol microphysical effects are likely to cause non-linear CCN variations. Whereas many modelling studies have assessed the main features of global variations in the aerosol particle size distribution (e.g. Ghan et al., 2001; Adams and Seinfeld, 2002; Spracklen et al., 2005) and several

have explored aerosol-cloud interactions in regional scale models (e.g. Bangert et al., 2011; Yang et al., 2012) only a few studies (Ekman et al, 2004; 2006; Wang et al., 2011; Archer-Nicholls et al., 2016) have explored the microphysical properties of aerosols, and their potential interactions with clouds, at resolutions of ~1km where convection is resolved.

It is known that deep convection can lead to transport of aerosols (e.g. Yin et al., 2012). In arid environments, cold-pool outflows from convection can be a major source of dust uplift, which is missed by large-scale models that parameterise moist

convection (Marsham et al., 2011; 2013; Pope et al., 2016). Similarly, it has been shown that over oceans such convectively generated flows can both increase gaseous DMS emission and transport, since the convection generates locally strong winds leading to high emissions that are then preferentially transported by the convection (Devine et al., 2006). There are, however, few model studies of aerosols in ocean environments with deep convection (e.g. Cui et al., 2011) or shallow convection (e.g. Kaufman et al., 2005).

The main objective of the current study is to assess spatial and temporal variations in aerosol properties in a convection-permitting resolution model (grid-spacing ~1 km), in particular investigating the concentration range of different sized CCN, considering potential implications for aerosol-cloud interactions simulated by current composition-climate models. In order to well characterize the influence of both the dynamics and aerosol microphysical influences on cloud-relevant aerosol properties, the GLOMAP aerosol microphysics scheme is applied at high resolution over an idealised three-dimensional

tropical marine domain. The convection-permitting aerosol microphysics simulations represent a highly realistic representation of CCN variations, providing a ground-breaking research tool for investigating aerosol-cloud interactions. The model includes interactive emissions of DMS and sea-spray and an online tropospheric chemistry scheme, ensuring the simulations include a comprehensive treatment of the combined effects from dynamical, chemical and aerosol-microphysics processes occurring in the marine boundary layer. The paper is organised as follows: after a description of the UKCA (UK

Chemistry and Aerosol) model and its high resolution configuration in Section 2, simulation results are described in Section 3. Section 4 summarizes, concludes and discusses the findings.





## 2 Model description

### 2.1 The UK Chemistry and Aerosol model (UKCA)

The UKCA sub-model of the UK Met Office Unified Model (MetUM) is used (hereafter UM-UKCA), including the GLOMAP-mode aerosol microphysics scheme (Mann et al., 2010) which calculates the evolution of aerosol mass and

number in several log-normal size modes. The scheme represents each size mode as an internal mixture, with several aerosol components able to be simulated including sulphate (SU), sea-salt (SS), dust (DU), black carbon (BC), and particulate organic matter (POM) (including primary and biogenic secondary POM). Any number of modes (with fixed standard deviation) and possible components can be tracked, but the simulations here apply the "standard" configuration used in UM-UKCA (e.g. as in Bellouin et al., 2013) with 4 components (SU, SS, BC, POM) in 5 modes (Table 1) and dust transported

separately in the existing 6-bin MetUM scheme (Woodward, 2001). The aerosol processes are simulated in a size-resolved manner and include primary emissions, secondary particle formation by binary homogeneous nucleation of sulphuric acid and water, growth by coagulation, condensation and cloud-processing and removal by dry deposition, nucleation scavenging, impaction scavenging and sedimentation.

The standard tropospheric chemistry configuration of UM-UKCA is used (O'Connor et al., 2014) which includes $O_x$-$HO_x$-

$NO_y$ chemistry with degradation of methane, ethane and propane. The implementation here also includes the extension for aerosol precursor chemistry (as in Bellouin et al., 2013) for the oxidation of sulphur precursors DMS and $SO_2$, and produces secondary organic aerosols via gas-phase oxidation of a biogenic monoterpene tracer.

### 2.2 High-resolution configuration of UM-UKCA

The simulations are carried out with UM-UKCA applied in a high resolution limited area model with periodic lateral

boundary conditions, specifically applying the Numerical Weather Prediction configuration of MetUM GA4.0 (Walters et al., 2014). MetUM GA4.0 provides tracer transport, boundary-layer mixing, large-scale cloud and precipitation, with UKCA simulating atmospheric chemistry and aerosol processes. The limited area domain is centred close to the equator (1.32°, 1.08°) and set to 240 km x 240 km with 1.5 km horizontal grid-spacing. At this resolution, much of the convective-scale dynamics is resolved, and the MetUM convection parameterization (Gregory and Rowntree, 1990) is not applied. Cloud

microphysics is represented using a single moment scheme (Wilson and Ballard, 1999). In these idealised simulations the radiation scheme was switched off, with the model therefore evolving without a diurnal cycle introduced by the daily variation in solar insolation or in variations in long wave cooling. The prognostic aerosols therefore do not affect clear-sky radiative transfer. The interactive CCN variations also do not feed through to modified cloud physics, with the investigation here exploring only variations in aerosol properties. A 73-level vertical grid is used up to a model top of 80 km, with 50

levels in the troposphere, 21 of which span the lowest 2 km of the atmosphere. Only a short demonstration simulation is here carried out with just 24 h integration time from an initial time of 09:00 UTC on May 24, 2002. Emissions are calculated on the simulation timestep of 30 s with UKCA chemistry and aerosol processes integrated every 10 timesteps, i.e. every 5 min.





Emissions of DMS and sea spray are interactive in the model, with their flux into the atmosphere primarily driven by variations in the model wind speed (using the same approaches described in Bellouin et al., 2013). Anthropogenic emissions of $SO_2$ and BC/OC are taken from the relevant grid-cell from the IPCC AR5 global emission data (Lamarque et al., 2010), with monoterpene and biomass BC/OC emissions from the GEIA[1] and GFEDv2[2] databases respectively, but sources from these sectors are not significant in this domain.

The thermodynamic (temperature and humidity) and dynamic (horizontal wind) variables are initialised from a single model profile (Figure 1) taken from a global aqua-planet configuration of a MetUM operational run (where all land points are removed). The profiles are deliberately chosen to be strongly unstable so that the model will experience a sudden deep convective instability in the early phase of its evolution. The convective perturbation can clearly be seen in Figure 2, with deep convective clouds forming after a few hours, reaching up to a cloud top height of approximately 18 km (Figure 2a). The system becomes precipitating after ~5 hours of simulation (Figure 2b) with the surface rain rate becoming more intense after 12 h and building up to a maximum of approximately 80 mm h$^{-1}$ between 14 and 16 h of simulation. The mean horizontal surface wind speeds over the domain increase only slightly from 2 up to around 4 m s$^{-1}$ as the storm develops (Figure 2c), but variations within the domain are large with a maximum one standard deviation range of 3 to 9 m s$^{-1}$ and strong wind speeds occurring consistently between 6 h and 10 h of simulation (15:00 and 19:00 UTC). Between 3 and 6 h of integration (12:00 to 15:00 UTC), intense vertical wind speeds occur (Figure 2d) and those upward movements will transport DMS into the free troposphere where, after oxidation, it is known to cause new aerosol particle formation (e.g. Raes et al., 1993), with subsequent growth and re-entrainment into the boundary layer of the resulting secondary particles constituting a major source of marine CCN on the global scale (e.g. Korhonen et al., 2008). The sharp variations in horizontal wind speeds will also induce strong variations in the emission of sea-spray particles, since their source function has a cubic dependence on horizontal wind speed (e.g. Gong et al., 2002). Figures 3a-c present the variation in aerosol particle concentrations across the domain at the time of maximum convective instability with squall lines and associated cold pooling clearly apparent, with very strong particle concentration gradients across the gust fronts, and the gravity currents inducing regions of greatly enhanced sea-spray emission. The strong convective event causes a rapid spin-up of the atmospheric composition in the model, giving an opportunity to assess the variation in aerosol properties across a range of wind speeds during the decay after the storm has subsided. In the next section, these high-resolution spatial variations in size-resolved aerosol properties are explored, examining how the different aerosol sources and processes represented in the simulations influence fluctuations in marine boundary layer CCN concentrations at this convection-permitting resolution.

---

[1] Global Emissions Inventory Activity: www.geiacenter.org
[2] Global Fire Emissions Database, Version2: www.globalfiredata.org



# 3 Results

## 3.1 Gas-to-particle conversion

To aid interpretation and inference from the assessment of aerosol properties in subsequent sections, in this first part of the results we explore how the substantial emission of DMS during the intense storm period propagates through to simulated

concentrations of its oxidised forms sulphur dioxide ($SO_2$) and sulphuric acid ($H_2SO_4$). Emissions of DMS vary strongly with wind speed and emissions fluxes will therefore be highest between 7 h and 9 h when the peak of the wind speed fluctuations is at maximum. The high emissions lead to a peak in the domain-mean DMS concentration with maximum of ~10 pptm after 9 h of simulations (i.e. at 18:00 UTC). DMS is oxidised by OH during the daytime and by $NO_3$ at night, both reactions producing $SO_2$ which, in the gas phase, goes on to form $H_2SO_4$ vapour following further reaction with OH. Figure

4 illustrates the timescales associated with these processes. The domain-averaged surface $SO_2$ and $H_2SO_4$ concentrations are peaking much later than DMS (22 h of simulation, 07:00 UTC) at respectively ~18 and 6.5 x $10^{-3}$ pptm. Given the photochemistry involved, the peak concentration at 07:00 UTC is surprising, but illustrates how atmospheric composition at the surface is strongly influenced by dynamical effects, not just atmospheric chemistry. That the model was still spinning up at this time is greatly beneficial as it helps identify which processes cause the CCN variations and allows to better understand

the temporal signatures of the different processes involved. The gas phase $H_2SO_4$ produced from the emitted DMS is a prerequisite for effective new particle formation and also causes growth of existing particles following vapour condensation, both effects being important sources of marine cloud condensation nuclei (e.g. Korhonen et al., 2008). Although BC and POM are resolved in the model, and UKCA chemistry includes the oxidation of monoterpenes, their emission in this marine domain is negligible. Rather sea spray and DMS-derived sulphate particles are the only two significant particle sources in

these simulations.

Hereafter, the analysis focuses on assessing separately the aerosol particles in the different size modes, investigating how the identified driver sources and processes are influencing simulated CCN variations at this convection permitting resolution. The analysis is restricted to the last 12 h of simulation with an emphasis on the results obtained after 18 h of integration, by which time the model has fully spun-up.

## 25 3.2 Properties of the aerosol fields

In this section, the focus is on quantifying variations in aerosol properties in the three different particle size ranges: Aitken, accumulation and coarse modes. The analysis begins (Figure 3) with instantaneous snapshots of surface aerosol particle concentration and size at two different times in the simulation. Figures 3a-c present a snapshot of spatial variability at 6 h of integration, when an intense storm period was occurring. Figures 3d-i show the snapshot spatial variation at 18 h of

integration, in more modest and representative wind speed conditions. The coarse mode consists entirely of sea-spray particles, so highest particle concentrations are expected to generally be indicating regions where simulated horizontal wind speeds are highest. However, during the initial storm period, and at this high spatial resolution, there are also regions of



intense localised precipitation and powerful vertical wind speeds, which will also strongly influence aerosol properties due to removal and transport effects. At 6 h of simulation, Figures 3a-c show that particle concentrations in the two largest modes (accumulation and coarse) are indeed extremely variable over the entire domain. For example, particle concentrations vary from 1 to 1,000 $cm^{-3}$ for the accumulation mode and from 0.1 to 100 $cm^{-3}$ for the coarse mode. Note however that this very high aerosol variability is unrealistically large, being mostly due to the model being initialised with a "warm bubble" to ensure model spin-up proceeds rapidly. However, the period from 12 to 24 hours of integration can be considered to span representative range of wind speed conditions, and we focus on this 2$^{nd}$ half of the day in the rest of the results sections.

In this remote marine domain, particles in the Aitken mode are almost exclusively secondary in nature, being originally formed via nucleation in the free troposphere. Over the initial 12 hours, free troposphere concentrations of the driver gas for nucleation, $H_2SO_4$ are not yet high enough to initiate significant particle formation, with low simulated concentrations of its precursor species $SO_2$ (see Figure 4) and timescales for oxidation and transport being relative long. After 18 h of simulations, the strongly convective episode has passed, and coarse mode particle concentrations (Figure 3f), although still quite variable, have more moderate peak concentrations, lower by around a factor of 10 than during the storm period (Figures 3c, f). Accumulation mode particle concentrations at 18 h (Figure 3e) are also much less variable than at 6 h, with highest concentration in the same regions that coarse mode particle concentrations were highest, likely indicating where sea spray emissions are highest (horizontal wind speeds are strongest). Patches of low concentrations are also found where the precipitation is most intense, with the washout rate (impaction scavenging efficiency) tied to rainfall rates. In the Aitken mode (Figure 3d), particle concentrations have become significant by 18 h, although still an order of magnitude lower than in the accumulation mode. Spatial variations in the size of the aerosol particles is also highest for the coarse mode (Figure 3i), with regions of highest particle concentration generally corresponding to smaller particles, likely reflecting the nature of the sea spray source function. The general spatial patterns of size variation seen for the coarse mode are also seen for the accumulation mode (Figure 3h) but the accumulation mode has additional regions of lower particle size where Aitken mode particle concentrations are highest (Figure 3d). This co-variation is expected, since the accumulation mode mean radius will be lower, on average, when there are a significant number of smaller particles being chemically cloud processed or mode-merged in from the Aitken mode. Over the domain, mean particle size variations are largest for the Aitken mode at 118% min-to-max ratio (geometric mean radius from 22 to 48 nm), compared to ~20% for the accumulation mode (101 to 123 nm) and ~35% for the coarse mode (0.75 to 1.10 μm).

In Figure 5 we show Hovmöller diagrams to further explore the temporal evolution in surface concentrations of Aitken, accumulation and coarse mode particles during the last 12 h of integration (at y = 150 km). Highest particle concentrations from accumulation and coarse modes are apparent between 12 and 15 h of integrations, whereas Aitken mode particle concentrations evolve with quite different time-variation. The convective storm period in the first 12 hours causes very strong wind speeds and the decay of the coarse mode particles concentrations over this second half of the day reflects the progression to calmer conditions, with consequently reduced sea spray emissions. By contrast, Aitken particle concentrations



are steadily increasing to a maximum of around 1-2 particles per $cm^3$ at 22 h, matching that seen for $SO_2$ and gas phase $H_2SO_4$ (Figure 4), consistent with the timescales of the two oxidation steps required to convert enough of the emitted DMS into sulphuric acid vapour to trigger new particle formation. For particles to reach Aitken sizes, growth by condensation and coagulation is also required, and since nucleation will mostly tend to occur in the free troposphere, any transition to a

statically stable boundary layer during late evening would likely also be important, influencing particle entrainment and the timing of the increase in Aitken particle concentrations at the surface. In these idealised simulations however, the short wave and long wave radiation schemes are switched off, there then being no solar-induced diurnal variations in boundary layer entrainment (but photochemical variations proceeding in the model based on local time).

Assessing how each of the size modes is spinning up reveals how temporal variations in marine CCN concentrations are
actually reflecting the very different time-profiles of the two dominant CCN production pathways: primary emissions of sea-spray particles and entrainment of DMS-derived secondary particles formed in the free troposphere. The analysis illustrates the way a diverse community of processes (dynamical, chemical and microphysical) together determine CCN variations in the marine boundary layer. Figure 5a shows an Aitken mode emerging after 17 h of integration which also explains the dip in accumulation mode size (contour lines), as a substantial number of smaller secondary particles is being "mode-merged in"
from the Aitken mode at that time. For the coarse mode, as particle concentrations decrease, there is also a progression to smaller particles, which can be explained by that fact that, in the model, sedimentation (the dominant removal process for this mode) removes both number and mass, enabling the simulation to reflect the fact that larger particles fall faster even when they are in the same mode.

A more quantitative analysis of the simulated aerosol properties is presented hereafter, with Figure 6 showing probability
density functions (PDFs) of the geometric dry radius (a-c) and particle concentrations (d-f) for the Aitken, accumulation and coarse modes at different times in the 2nd twelve hours of the integration. The analysis shows that, for the accumulation and coarse modes, as seen in Figure 5, as time progresses, the particle size PDFs shift to smaller sizes, with the accumulation mode PDFs becoming much wider in the evening as the source of smaller particles from the Aitken mode becomes significant. By contrast, as Aitken mode concentrations increase, the particles are clearly also larger, reflecting that growth
processes are acting on the particles with this size-increase ceasing at about 18 h, while particle concentrations continue to increase (likely due to entrainment). For the accumulation and coarse mode particles, this quantitative approach is consistent with sedimentation causing the shift in size distribution as the larger particles sediment out faster than the smaller ones. Figure 7 shows the temporal evolution of the mean and standard deviation of the geometric mean radius values and number concentration (over grid boxes in the domain) for Aitken, accumulation and coarse modes at the surface. The accumulation
and coarse mode concentration and radius fields have largest spatial variations between 5 and 8 h as the model adjusts to the very strong sea-salt emission and wet removal during the peak convective activity, whereas Aitken mode concentrations, and their variations, stay approximately constant through that period. During the simulations, the mean radius and particle concentration values from the coarse mode are, on average, decreasing (Figures 7c, f), but the mean size variations show the





opposite evolution, with greater variability in the calmer 2nd half of the day, reflecting the strengthening influence of sedimentation as sea-spray emissions decrease. For the accumulation mode, the mean particle size displays remarkably little variation over the domain between 9 and 14 h of simulation (as seen in Figure 6a), with the variation increasing as the source of secondary CCN from the Aitken mode becomes significant later in the day.

**3.3 CCN spatial and temporal features**

In this marine domain, sea-salt particles represent a major component of the CCN population (e.g. O'Dowd et al., 1997; O'Dowd and de Leeuw, 2007). Models parameterize sea-spray emission fluxes as a function of the 10 m wind speed ($u_{10}$), with some source functions linked directly to field measurements of the particle concentrations (e.g. Smith et al., 1993) while others (e.g. Monahan et al., 1986) reflect also the processes that form ocean whitecaps, and laboratory experiments on

particle emissions. In the simulations presented here, the model uses the sea spray source function of Gong (2003) which applies the approach of Monahan et al. (1986), and its $u^{3.41}$ 10 m-wind-speed dependence, with a refined formulation with an additional parameter determining emission of ultra-fine sea-spray particles, as constrained by field measurements from O'Dowd et al. (1997).

In light of the inference of sea-spray emissions fluxes from measurements of particle concentrations, Figure 8 presents

several snapshot variation box-plots for simulated sea-salt emission flux, sea-salt mass mixing ratio (mmr) and the CCN number concentration as a function of the $u_{10}$ (called hereafter surface wind speed) at different integration times. At this convection-permitting resolution, the sea-salt emission fields are highly heterogeneous at each integration time with the emission flux median highest at 12 h of integration then decreasing towards the end of the simulations, consistent with the mean wind speed evolution (Figure 2c). After their emission into the atmosphere, sea-salt aerosols are transported vertically

by turbulence, with larger particles also being influenced by sedimentation. As expected, near the surface, the higher are the sea-salt emission fluxes, the higher are the sea-salt mmr, but, as we show below, the co-variation of the sea-salt mmr field with wind speed (Figure 8b) is fundamentally different than it is for sea-salt emission (Figure 8a). Sea-spray particles are highly soluble and are, in most cases, directly emitted at sizes where they are effective CCN, but, as discussed earlier, in marine regions, the CCN population also has a substantial contribution from nucleated sulphate particles which have grown

large enough to be CCN-active. Figure 8c shows the variation of CCN concentrations in this marine domain at different integration times and permits to explore how its variation compares to that seen for sea-spray mmr and emissions flux. By sampling the four periods at 12, 15, 18 and 21 h, it is possible to assess the spatial variability in sea-salt emissions, sea-salt concentrations, and CCN concentrations at a range of wind speed conditions; the earliest period representing a strong convective period when sea-spray would be dominant, and through the progression to calmer conditions later in the

simulation. First, the relative change in the median between each period (12-15, 15-18 and 18-21 h) is assessed. As expected, the median sea-salt emission flux (Figure 8a) decreases linearly on the log-log plot over the period, reflecting the 3.41 exponent in the wind-speed-dependence for the sea-spray source function. The relative decrease in sea-spray emissions





between 12 h and 15 h is reflected also between 15 h and 18 h, and between 18 h and 21 h. By contrast, the median sea-salt mixing ratio (Figure 8b) decreases much more steeply over the 18-21 h period than in the 12-15 h period, despite strong decreases in wind speed. This effect is likely a result of the timescale for the decay from the excited state acquired during the strong convection period (the strong turbulence and direct transport having lifted particles much higher) the atmosphere still "catching up", with the adjustment to the new calmer conditions only being visible after 18 h. The equivalent temporal decay for CCN concentrations is also curved (Figure 8c), but part of the signal of steeper decline between 18 and 21 h (from the decreased sea-salt) is "straightened out" by the compensating emergence of the secondary nucleated particles making an important contribution to CCN in this later period (as we showed in Figure 7). In the calm evening conditions, wind speeds across the domain vary ($5^{th}$ to $95^{th}$ percentile) from ~0.21 to 1.6 m $s^{-1}$, around a factor of 8, with the CCN concentration range from around 6 to 20 particles per cubic centimetre (a factor of 3). In contrast, during the strongly convective period, the CCN variation is much larger, between 12 and 95 particles per cubic centimetre (a factor of 8).

Figure 9 presents the vertical variation of the simulated CCN concentration using an altitude PDF profile (a-PDF) for the same periods as mentioned above. As expected, on average, the CCN concentration drops-off with increasing altitude reflecting a balance between turbulence and convection lifting the particles vertically and gravitational settling transporting larger particles back towards the surface. In the atmospheric surface layer (lowest 100 metres or so) the profile of mean CCN follows a power-law profile but the spatial CCN variance (standard deviation over grid boxes in the domain) decreases much less rapidly with altitude. As a consequence, the coefficient of variation increases with increasing altitude from approx. 11% at the surface to approx. 22.5% at 1.3 km height. The CCN concentration fields close to the surface are mainly influenced by the emissions whereas at higher altitudes they are mostly influenced by the transport. This explains why, after 12 h of simulation, the coefficient of variation is slightly higher than at the others times. Note that the emergence of the secondary nucleated particles is also visible in the CCN concentration vertical properties on the 18-21 h period.

## 4 Conclusions and Discussions

We have analysed spatial and temporal sea-spray and CCN variations in a convection-permitting model with interactive sea-spray emissions, sulphur chemistry and aerosol microphysics over an idealised marine tropical domain. In this marine atmosphere the two dominant CCN sources are both natural – the cloud nuclei population comprising two elements – primary sea-spray particles and secondary sulphate particles. However, even in this relatively simple two-component CCN system, our analysis has revealed that there is a diverse community of processes: dynamical, chemical, and microphysical, that together combine to determine the number of particles which can activate to cloud droplets.

First, the dynamics strongly influences the sea-spray emissions since highest particle concentrations occur where wind speeds are highest, and there is a cubic wind speed dependence for sea-salt emission. The emitted sea-spray particles have a range of sizes, being directly emitted in both the accumulation (sub-micron) and coarse (super-micron) modes. After their emission into the atmosphere, sea-salt aerosols are transported vertically by turbulent diffusion, with larger particles also





being influenced by sedimentation. We show that the co-variation of sea-salt mass mixing ratio with wind speed is fundamentally different than that for sea-salt emission, with implications for derivations that treat the two synonymously. In particular, since sub-micron sea-spray has much longer atmospheric residence time (days) than super-micron sea-spray (hours), care must be taken when relating measured sea-spray concentrations to emissions. Intense localised precipitation

during strong convection also impacts aerosol concentrations at the climate grid-scale with removal effects introducing strong variations (e.g. via the impaction scavenging process). The combination of these processes impacts the particle concentration properties, which become extremely variable in space (about a factor of 8 over the entire domain, one climate model grid square) and time.

Moreover, the emissions of DMS strongly vary spatially and temporally according to wind speeds conditions and become

substantial during intense storm period (as in Devine et al., 2006). There is a requirement for gas phase species $SO_2$ and $H_2SO_4$ vapour to be sufficiently produced following oxidation of DMS before new sulphate particle formation in the free troposphere can occur, and the latter species also cause enhanced growth of existing particles following condensation. The combination of the two oxidation steps being required to convert emitted DMS into sulphuric acid vapour, with also the timescales inherent in particle growth processes (e.g. coagulation and condensation), explain why here is a quite different

time-variation for the Aitken mode particle concentrations. Provided the airmass has had sufficient time, a significant proportion of these small secondary particles grow large enough to be cloud processed or mode-merged from the Aitken mode to the accumulation mode. The effects of these processes is illuminated by assessing how each of the particle modes is spinning up, revealing the way they influence spatial- and temporal CCN variations in the marine boundary layer.

Sea-spray particles are highly soluble and, in most cases, are directly emitted at sizes where they are already effective CCN.

In contrast, a different component of the CCN population comprises nucleated sulphate particles which need more time to grow large enough to be CCN-active. The variations in the CCN concentrations are strong and can attain a factor of 8 in strongly convective conditions, mostly reflecting the properties of larger CCN. Smaller (sub-micron) CCN, from the accumulation mode, tend to have less variation, which in part is due to their source having a significant contribution from the steady formation of secondary sulphate particles in the free troposphere. We have seen how dynamics and microphysical

processes also affect CCN, in particular with a $2^{nd}$ CCN peak at the top of the boundary layer during the strongly convective period before the secondary particles emerged. These effects combine to determine how the coefficient of variation in CCN concentration changes with altitude, our results suggesting an increase from around 10% at the surface to more than 20% at the top of the marine boundary layer. Whereas CCN concentration fields close to the surface are mainly influenced by the emissions, at higher altitudes they are in general older, and inheriting influences propagated via transport.

We also examine spatial and temporal variations in aerosol particle size, finding that the geometric radius of the Aitken and coarse modes are particularly variable, which will introduce further variability in cloud droplet number concentrations and cloud brightness. The different influences on the two CCN types (primary and secondary), and the diverse community of processes involved (microphysical, chemical and dynamical) makes sub-grid parameterization of the CCN variations difficult. This study provides valuable results on e.g. the impact of the local dynamics and aerosol sources on the CCN





population and then on the aerosol-cloud interactions occurring at these fine spatial scales. Work to apply the UM-UKCA model for non-idealised case-studies with a nesting procedure to retain the larger scale influences has now been developed, as is the capability to allow these aerosol variations to couple with a new cloud microphysics scheme in MetUM (Shipway and Hill, 2012).

*Acknowledgement.* This project was made possible through the financial support of the Leeds-Met Office Academic Partnership (ASCI project). The authors acknowledge use of the MONSooN system, a collaborative facility supplied under the Joint Weather and Climate Research Programme, which is a strategic partnership between the Met Office and the Natural Environment Research Council. The lead author wishes to thank Douglas Parker for useful discussions. G. W. Mann is

10    funded by the UK National Centre for Atmospheric Science, one of the UK Natural Environment Research Council (NERC) research centres. J. Marsham acknowledges funding from the SAMMBA project (NE/J009822/1).



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



**Table 1.** Standard aerosol configuration for GLOMAP-mode. The size distribution is described by lognormal modes with varying geometric mean diameter $D$ and fixed geometric standard deviation $\sigma_g$. Particle number and mass are transferred between modes when $D$ exceeds the upper limit for the mode. Names are given in function of the aerosols mode ('nuc', 'Ait', 'acc' and 'coa' are for 'nucleation', 'Aitken', 'accumulation' and 'coarse') and their solubility properties ('sol' and 'ins' mean the aerosols are soluble or insoluble). The aerosols can be composed of sulphate (SU), primary organic matter (POM), black carbon (BC), or sea-salt (SS).

| Index | Name | Size range | Composition | Soluble | $\sigma_g$ |
|-------|------|------------|-------------|---------|------------|
| 1 | nucsol | $D < 10$ nm | SU, POM | yes | 1.59 |
| 2 | Aitsol | $10$ nm $< D < 100$ nm | SU, BC, POM | yes | 1.59 |
| 3 | accsol | $100$ nm $< D < 1$ μm | SU, BC, POM, SS | yes | 1.59 |
| 4 | coasol | $D > 1$ μm | SU, BC, POM, SS | yes | 2.00 |
| 5 | Aitins | $10$ nm $< D < 100$ nm | BC, POM | no | 1.59 |





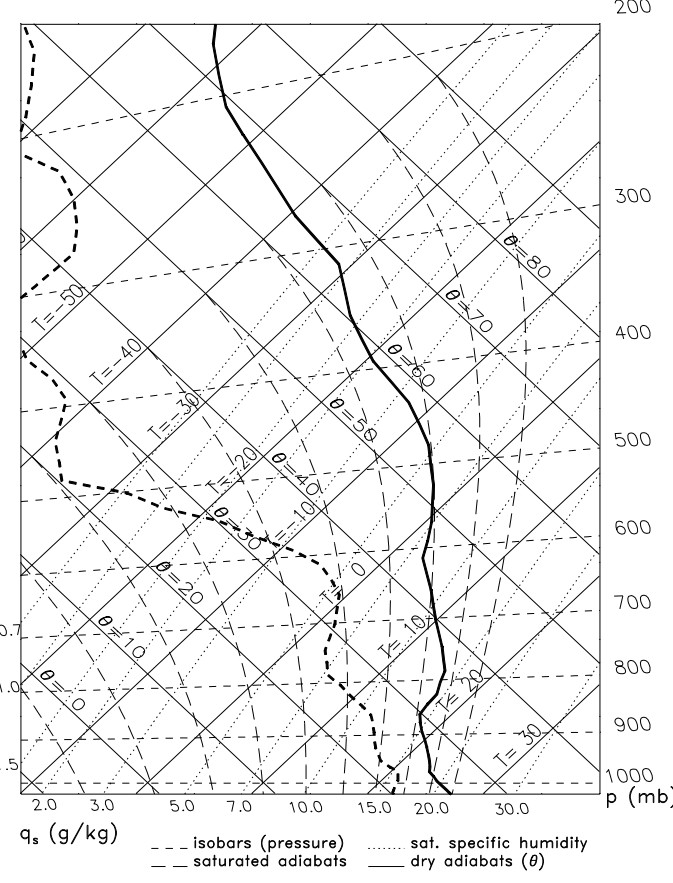

**Figure 1.** Tephigram representing the vertical profile of the initial dew point temperature (dashed line) and the temperature (solid line).





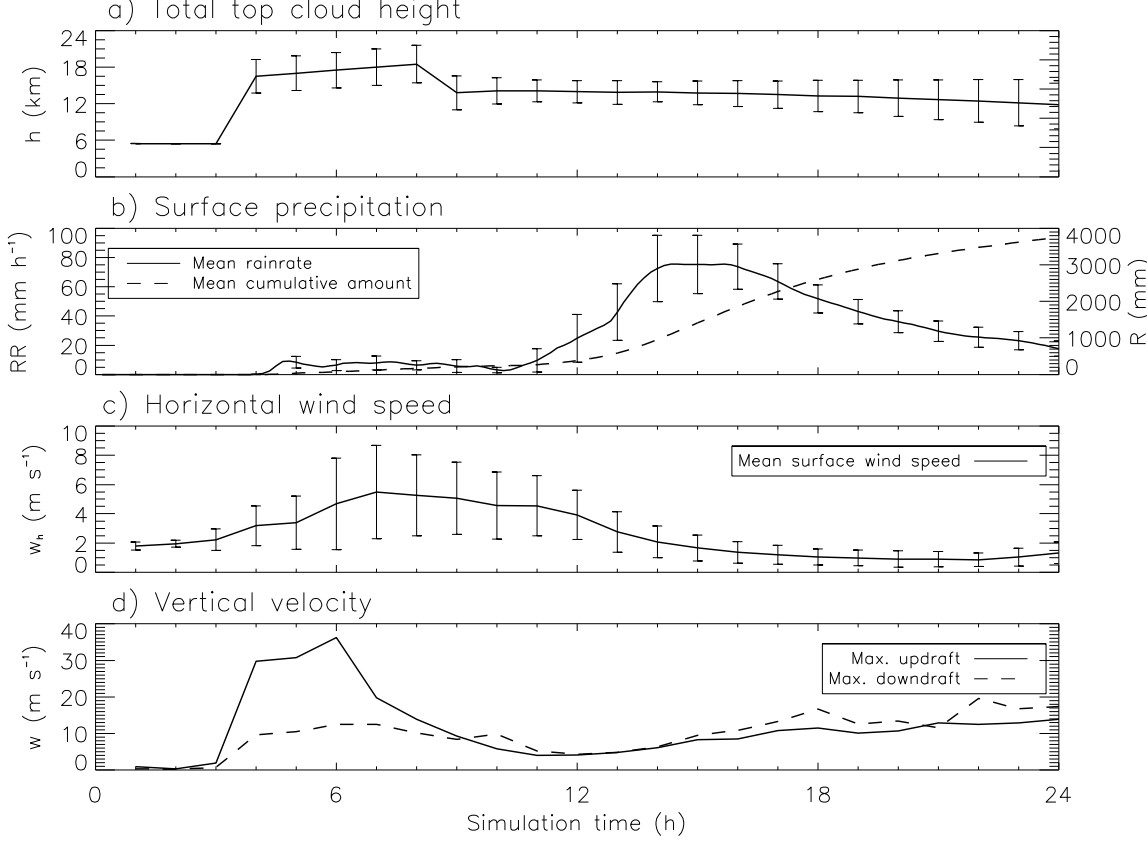

**Figure 2.** Temporal evolution of (a) the mean total top cloud height, (b) the mean rain accumulation and rain rate at the surface, (c) the mean surface horizontal wind speed and (d) the maximum of the updrafts and downdrafts. The averages are obtained over the entire grid points of the domain and given with ± one standard deviation.







**Figure 3.** Snapshot spatial variations in the number concentrations (a-f) and geometric-mean radius (g-i) of the aerosol particles in the Aitken (aitsol; a, d, g), accumulation (accsol; b, e, h) and coarse (coasol; c, f, i) soluble modes after 6 h (in model spin-up) (a-c) and 18 h (d-i) of integration. The black solid lines represent the surface vertical wind speed (w = 5 m s$^{-1}$). Note that the colour scales are different. The dashed lines correspond to the transects shown in Figures 5 and 9.





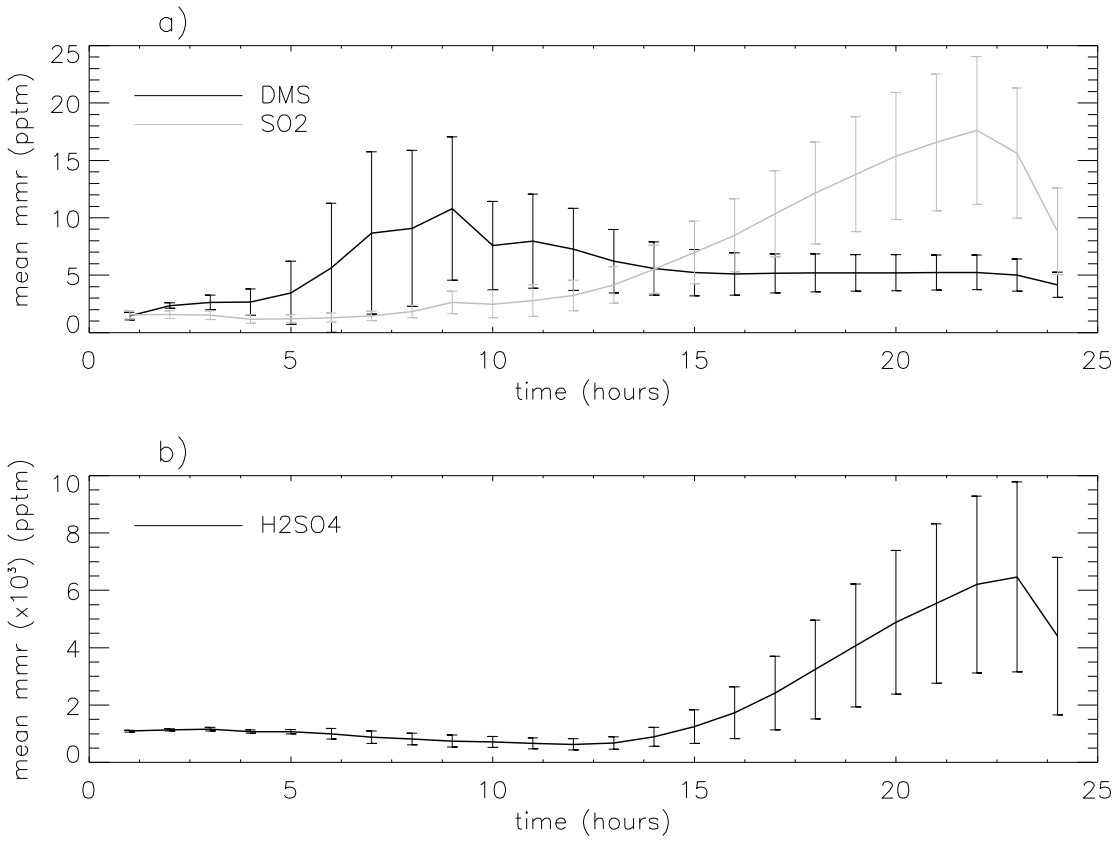

**Figure 4.** Temporal evolution of the mean mass mixing ratios of the gas precursors to aerosols. The DMS, $SO_2$ and $H_2SO_4$ mass concentrations are in pptm (part per trillion in mass). The error bars correspond to ± one standard deviation.



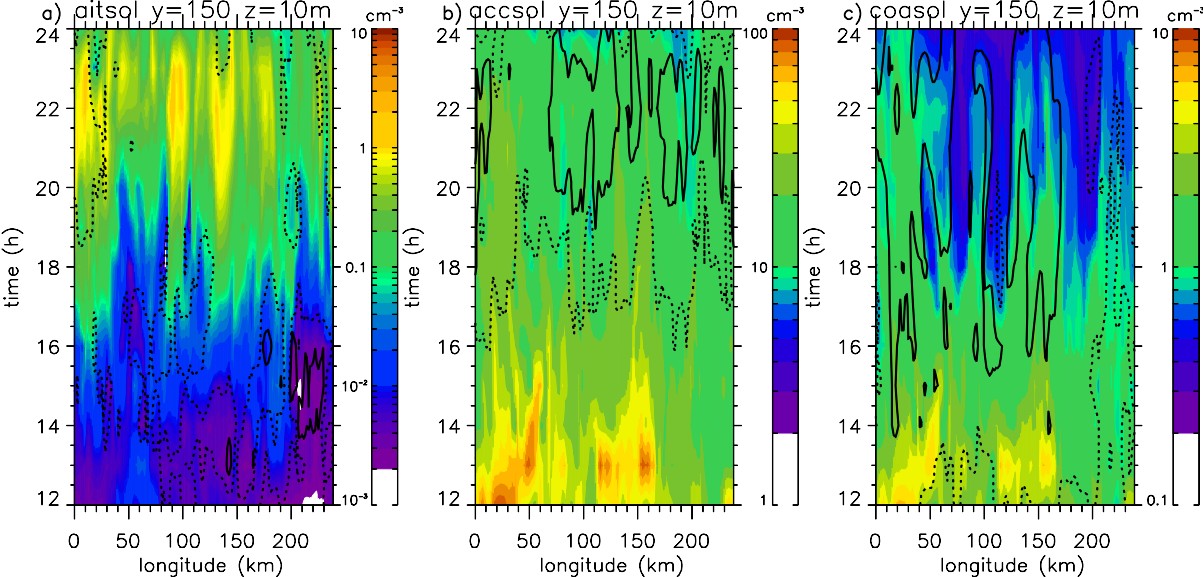

**Figure 5.** Temporal evolution of the aerosol concentration from the Aitken (a), accumulation (b) and coarse (c) soluble mode after the model spin-up. Temporal evolution of the aerosol dry radius is also illustrated for the 3 modes: 30 (dashed line) and 35 nm (solid line) for the Aiken soluble mode, 110 (solid line) and 115 nm (dashed line) for the accumulation soluble mode; and 0.96 (solid line) and 1.0 μm (dashed lines) for coarse soluble mode. Note that the colour scales are different.





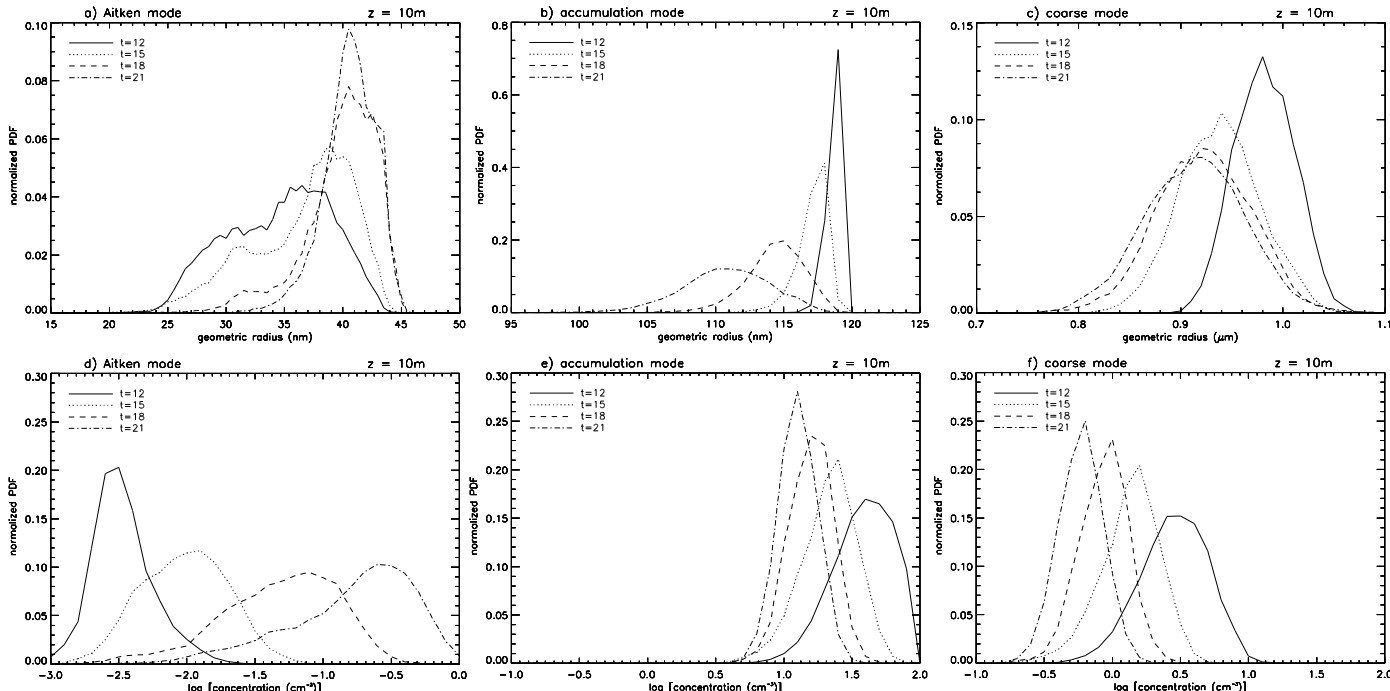

**Figure 6.** Normalized Probability Density Function (PDF) of the geometric radius (a, b, c) and the logarithm of the concentration (d, e, f) of the surface aerosols from the Aitken (a, d), accumulation (b, e) and coarse (c, f) soluble modes obtained at different integration time.



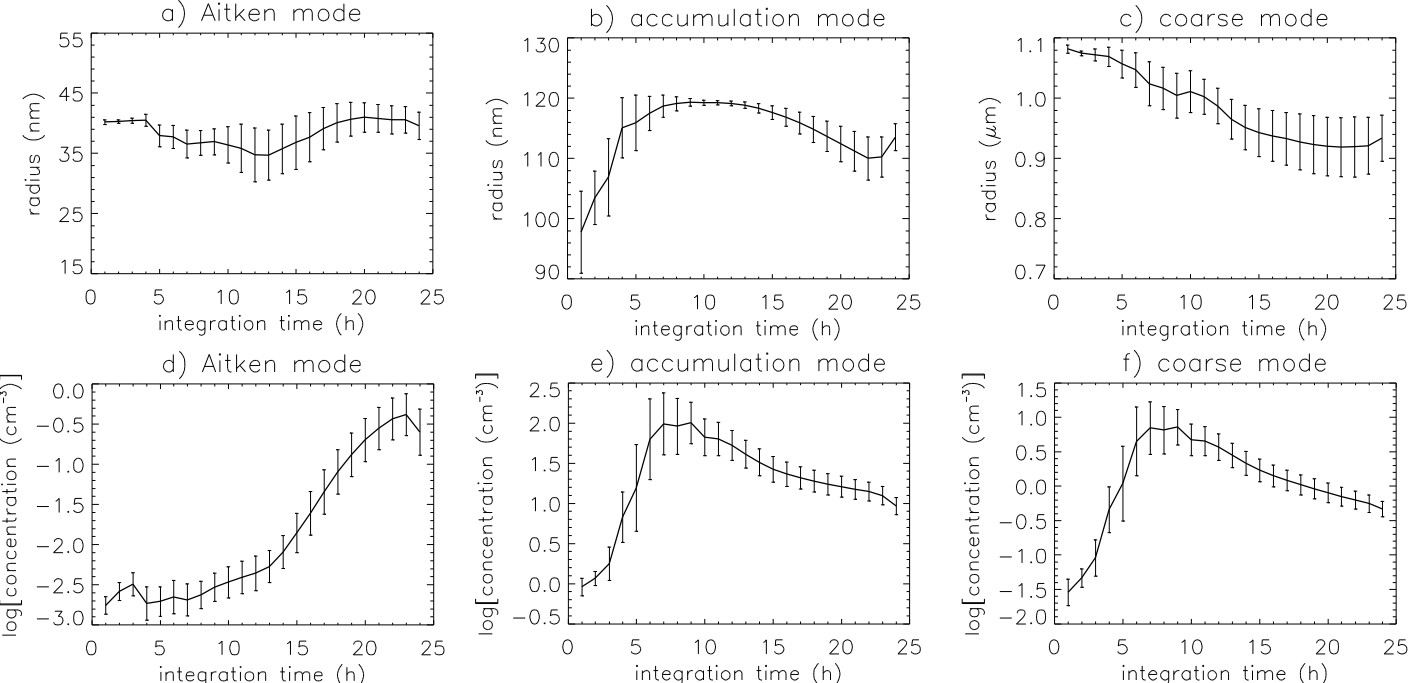

**Figure 7.** Time series of the mean ± one standard deviation of the geometric radius (a, b, c) and the logarithm of the concentration (d, e, f) of the surface aerosols from the Aitken (a, d), accumulation (b, e) and coarse (c, f) soluble modes.





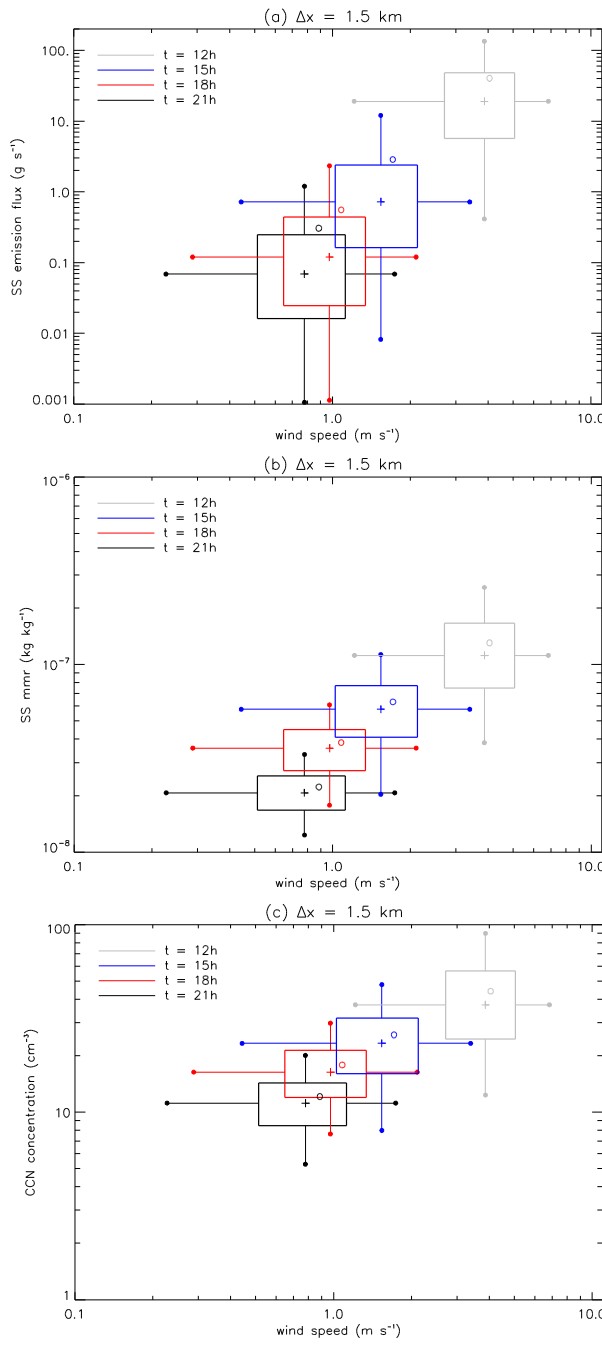

**Figure 8.** 2D-distribution of the surface sea-salt emission flux (a), sea-salt mass mixing ratio (mmr) (b) and CCN concentration (c) as a function of the surface horizontal wind for 4 different integration times (t = 12, 15, 18 or 21h). The hinges of the box-plots represent the 25[th] and 75[th] percentiles and the ends of the whiskers (full circles) represent the 5[th] and 95[th] percentiles. The 'plus' symbols represent the median values. The empty circles show the mean values over the domain.



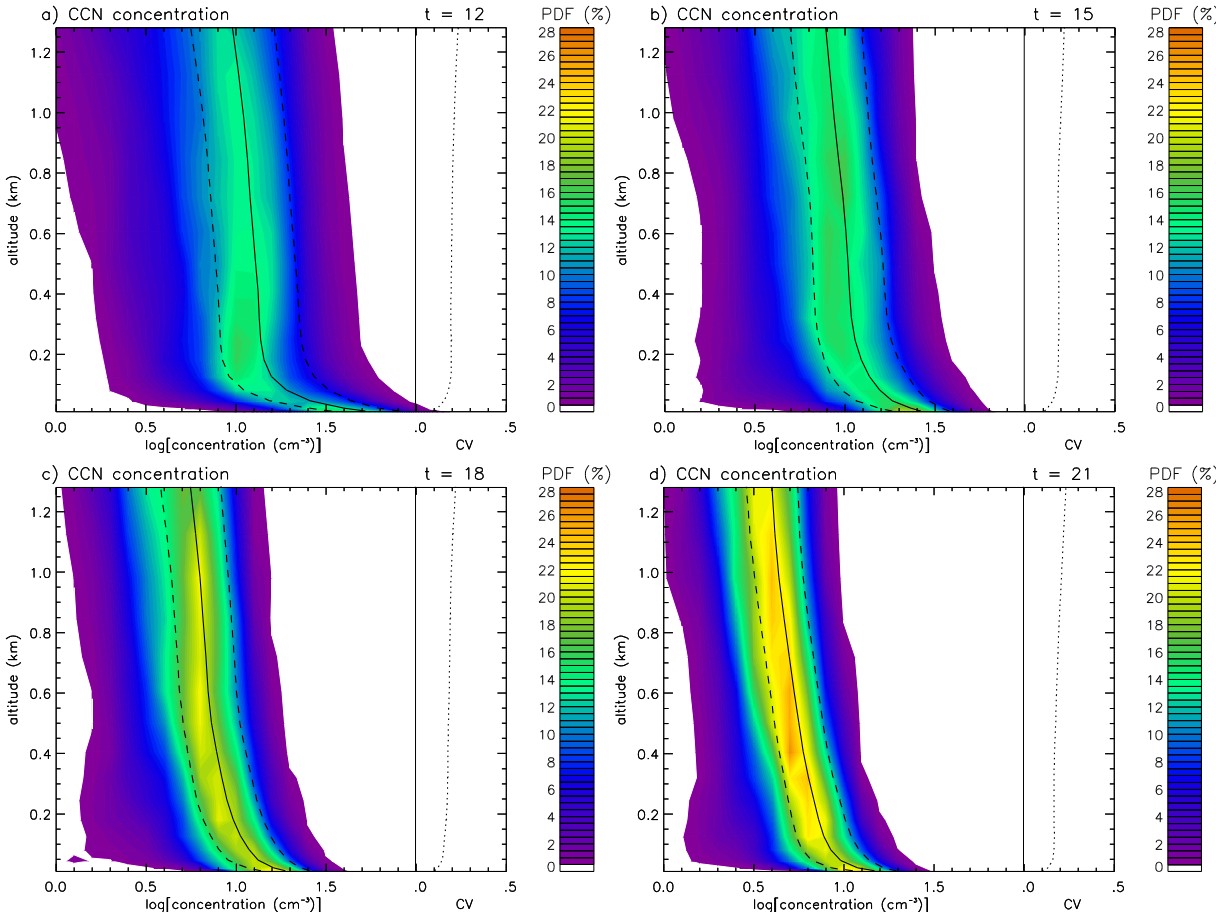

**Figure 9.** Altitude-dependent Probability Density Function (a-PDF) in percent of the CCN concentration at different integration times. The a-PDF are obtained calculating the PDF for each different level. A resolution of 0.1 is used for quantify the logarithm of the concentration. The lines represent the mean (solid lines) ± one standard deviation (dashed lines) of the CCN concentration. The dotted lines represent the coefficient of variation (CV) which is defined as the ratio of the standard deviation to the mean of the CCN concentration.