# Peer review of "Spatial and temporal CCN variations in convection-permitting aerosol microphysics simulations in an idealised marine tropical domain"

_Atmospheric Chemistry and Physics, 2016_

## Referee Comment (RC1) · Anonymous Referee #1 · 7 Nov 2016

**Review for "Spatial and temporal CCN variations in convection-permitting aerosol microphysics simulations in an idealised marine tropical domain"**

Paper summary and recommendation:

This paper disentangles the contribution of different processes to the overall CCN variability detected over a domain the size of a conventional general circulation model (GCM) grid box for the case of a convective tropical marine boundary layer. The study is performed in a simplified idealised setup. Feedback pathways between aerosol concentrations and the environment via radiative or cloud microphyiscal interactions are ignored. Thereby, an attribution of different processes to CCN variability due to spatial and temporal variability of size and number of the 3 mixed-modes contributing to the CCN budget (Aitken, accumulation and coarse mode) is obtained. The authors show that CCN concentrations may vary up to a factor 3-8 throughout the simulation domain. Understanding the origins of this variability is an important step towards estimating the potential biases of aerosol-cloud interaction estimates obtained by GCMs, which do not resolve this variability. I therefore recommend this article for publication in Atmos. Chem. Phys. following minor revisions.

Minor Comments – general:

- I believe that your aerosol concentrations are spun up from an entirely clean (i.e. Naero=0.0 cm-3) atmosphere. Please state this explicitly in the manuscript. I agree with the authors that this gives you the opportunity to disentangle the individual processes. However, this may be at least partially responsible for the high variability in CCN (800%) obtained after 12h of simulation following the period of intensive updrafts. If that is the case, context should be provided for the interpretation of this estimate. If you initialised a homogeneous profile of e.g. accumulation mode aerosol, would you still obtain such a high degree of variability of CCN following the intense updraft period? Please comment.

- The phrase "strongly convective" period (or conditions) seems to refer to different things throughout the manuscript. Sometimes the phrase seems to be used to refer to the time period of intensive updrafts and strong horizontal winds and sometimes to periods of intense rain fall. Please define this term and use it consistently throughout the manuscript.

- It has been shown (e.g. Textor et al, 2006: "Analysis and quantification of the diversities of aerosol life cycles within AeroCom") that different assumptions made in modeling the sea salt flux may yield vastly different estimates of sea salt emission fluxes. How sensitive do the authors think their results are to their implemented SS emission parameterisation? Please comment.

Minor Comments – specific:

- P3L10: Please include reference Zubler et al (2011):"Simulation of dimming and brightening in Europe from 1958 to 2001 using a regional climate model", JGR, doi:10.1029/2010JD015396.
- P3L10-L12: Two recent studies have investigated the impact of resolution on aerosol variability and aerosol-cloud interactions in regional climate models down to the kilometre scale for boundary layer clouds. These references should be added:
    - Possner et al (2016): "The resolution dependence of cloud effects and ship-induced aerosol-cloud interactions in marine stratocumulus", JGR, doi:10.1002/2015JD024685.

- Weigum et al (2016): "Effect of aerosol subgrid variability on aerosol optical depth and cloud condensation nuclei: implications for global aerosol modelling", ACP doi:10.5194/acp-16-13619-2016.
- P3L18: Please clarify complexity of aerosol treatment here, as there have been numerous studies investigating the sensitivity of marine deep and shallow convection to simplified aerosol treatments.
- P3L23: Please rephrase "to well characterize".
- P3L25: How do the authors determine the "realistic" level of variability? Please add references here.
- P4L30: Please rephrase "only a short demonstration simulation is here carried out".
- P4L31: Please rephrase "carried on".
- P5L11: Please rephrase "becomes precipitating", "becoming more intense".
- P5L22: Please rephrase "associated cold pooling"
- P6L24: By which criterion do you define your simulation to have fully spun up? Please clarify.
- P7L20: The correspondence between patterns in highest particle concentrations and smaller particle sizes in Fig. 3 is not obvious to me in this particular figure. Please elaborate, or remove comment.
- P7L31ff: The second half of the day is not only characterised by calmer wind conditions, but also by intense precipitation between 12 – 18 h. I believe that it should be mentioned here.
- P8L31: "adjusts to the very strong sea-salt emission and wet removal". However precipitation only really intensifies much later than 8h after initialisation. Please comment on the role of wet removal during this period.
- P9L31 – P10L1: Remove sentence "The relative decrease in …". You already stated that it is linear.
- P10L30: "… sea-salt aerosol are transported vertically by turbulent diffusion", I would have thought that the convective updrafts would also contribute? Please comment, or adapt text.
- P11L9: Please rephrase "wind speeds condition".
- P11L21ff: The authors state that CCN variations can be as large as factor 8. This number is obtained 12h after the simulation (Fig. 8). At this time the winds subside and precipitation builds up. So, how well does it characterise the CCN variability obtained during the period of intense updrafts? It may be helpful to include a box diagram for 6h after initialisation in Fig. 8. Furthermore the authors state that the CCN variability is large whilst the accumulation mode variability is smaller. This is confusing as I would assume most CCN to stem from the accumulation mode (see Fig 7.). Please clarify.
- Fig 1: For illustrative purposes the authors may consider adding a line of adiabatic parcel ascent.
- Fig 2: Please rephrase "mean total top cloud height" to "mean cloud top height". Please rephrase "rain accumulation" to "accumulated rain" or "accumulated precipitation".
- Fig 4: For clarification it may help adding day and night markers for the sulfate chemistry.
- Fig6 and Fig7: Does your aerosol scheme specify modal boundaries for the Aitken, accumulation and coarse mode? If so what are these? These could be added in the model description section.
- Fig 7: What causes the large variability in radius for the accumulation mode particles up to 8h after initialisation? This is discussed in the text on P8L30ff, but I would have thought that the SS emission radius would be tighter constrained and that wet removal processes play a larger role later during the simulation (after 12h) as the RR peaks. Please clarify.

---

## Referee Comment (RC2) · Anonymous Referee #2 · 4 Dec 2016

**Review for: "Spatial and temporal CCN variations in convection-permitting aerosol microphysics simulations in an idealised marine tropical domain"**

**Summary and recommendation:**

This article employs a convection-permitting resolution model to assess the contribution of spatial and temporal variations in aerosol properties for the case of a convective tropical marine boundary layer to CCN variability across a domain the size of a GCM grid box. The model is setup in a simplified idealised configuration in which the radiation scheme was turned off and CCN concentrations do not feed through to the cloud microphysics. Subsequently, the current setup ignores feedbacks associated with aerosol-radiative and aerosol-cloud microphysical interactions that may impact the simulated aerosol field in the model. The authors find that the simulated CCN concentrations can vary significantly over the domain, more than a factor of 8 during strongly convective conditions. They assess the contribution of dynamical, chemical and microphysical processes to this high variability in CCN and attribute it to increased sea salt/DMS emissions when spatial and temporal wind speed fluctuations become resolved at this convection-permitting resolution, increasing peak wind-speeds. This is an interesting finding as current GCMs cannot explicitly resolve sub-grid scale variability in wind speeds. Such modelling frameworks are required to elicit the impact of spatial/temporal resolution in GCMs on the representation of aerosol-cloud interactions. Therefore, I recommend publication of this article ACP once the following revisions have been addressed:

**General comments:**

- The modelling framework developed is described as ground-breaking. One of the key advantages of the model is in the use of a unified modelling (UM) framework to investigate the dependence of parameters involved in aerosol-cloud interactions on model resolution. However, this strength has not been captivated upon in this study. There is a lack of evaluation of the impact of the increased resolution in the model on the parameters of interest. A comparison of the domain averaged parameter values presented in the study to the same parameters simulated by the GCM would be highly beneficial and greatly strengthen the conclusions presented. Does the observed sub-grid scale variability in CCN impact the average CCN concentration across the domain compared to a GCM? This comparison should be provided before publication in ACP.

- CCN represent the aerosol particles that can form cloud droplets under reasonable atmospheric supersaturations. Accordingly, CCN concentrations always refer to a specific supersaturation, for example, CCN (0.1%) or CCN (0.5%) and one should be careful when comparing CCN concentrations measured or simulated at different supersaturations. What supersaturation was used throughout the article for the CCN concentrations presented?

The variability in CCN concentrations reported in this idealised configuration has been shown to be strongly dependant on variability in wind speed across the domain. This is unsurprising considering the strong wind-speed dependence of the sea spray emission parameterisation employed. Accordingly more discussion is required as to the sensitivity of the results presented on the choice of sea spray emission parameterisation with regard to the following:

- As the findings presented are strongly linked to the simulated wind speed field across the domain some discussion is required as to how accurate the simulated wind field and convective perturbation is compared to the real world. Also, is the aerosol, thus, CCN

variability simulated expected compared to observations? Please discuss in relation to the footprint of flux measurements performed to measure sea-salt emissions in the marine environment and associated variability observed from these measurement campaigns.

- Numerous sea-salt emission parameterisations exist, derived from a variety of in-situ measurement campaigns and laboratory experiments. How does the chosen parameterisations wind-speed dependence compare to the range of parameterisations in the literature, e.g. Fig. 5 Salter et al., 2015? How might a different parameterisation alter the high variability in CCN across the domain found?

- The onset of wave breaking is important for sea spray aerosol formation. It is generally recognized that the whitecap fraction and therefore sea spray aerosol production is zero for wind speeds less than ~ 3 m s$^{-1}$ (Blanchard, 1963; Monahan, 1971). The implication of this with respect to the findings requires discussion, for example, what is the contribution of the total CCN variability simulated between 0-3 ms$^{-1}$? At what wind speeds does the CCN concentration begin to increase sharply, is there a threshold value?

- Discussion is required on the applicability of the chosen parameterisation of the resolution of the model (1Km) and time-step. Typically sea salt emission parameterisations are applicable to certain footprints, and parameterisations developed from in-situ observations are dependent on the memory of the wave field (a rising sea will result in a different emission profile than a falling sea). In addition parameterisations are developed using longer time windows for averaging for flux measurements compared to the model time-step employed. Is the sea spray source function being applied in the model at this temporal/spatial resolution in the way it was designed?

**Minor comments:**

- Section 2: A figure of the modified sea-spray source function used in the study would be beneficial here, especially for experimentalists.
- Section 2: For a modelling framework described as ground-breaking the model description is relatively sparse, for instance, how is hygroscopic growth parameterised in the model? This will affect the evolution of the aerosol field across the domain. Please provide a more detailed description of the aerosol microphysics scheme.
- Section 2.2: It is widely known 1-moment cloud microphysics schemes introduce errors compared to 2 or3 moment schemes. Some justification of this choice is required, was it due to computational restraints?
- Section 3.2, line 8: "Aitken mode are almost exclusively secondary in nature": Please reword, this is two strong, studies exist which show emission of sea spray in this size regime, e.g. Salter et al., 2015.
- Section 4, line 25. "comprising two elements": reword
- Fig.4: Why do the error bars in DMS & SO2/H2SO4 not correspond? Some discussion on expected oxidation timescales required, why is there no offset between H2SO4 & DMS observed?
- Recent studies have probed the dependence of aerosol processes on model resolution, for instance Weigum et al., 2016. This should be referenced.

Blanchard, D. C.: The electrification of the atmosphere by particles from bubbles in the sea, Prog. Oceanogr., 1, 171–202, 1963.

Monahan, E. C.: Oceanic whitecaps, J. Phys. Oceanogr., 1, 139– 144, 1971.

Salter, M. E., Zieger, P., Acosta Navarro, J. C., Grythe, H., Kirkevåg, A., Rosati, B., Riipinen, I., and Nilsson, E. D.: An empirically derived inorganic sea spray source function incorporating sea surface temperature, Atmos. Chem. Phys., 15, 11047-11066, doi:10.5194/acp-15-11047-2015, 2015.

Weigum et al.: Effect of aerosol subgrid variability on aerosol optical depth and cloud condensation nuclei: implications for global aerosol modelling, ACP doi:10.5194/acp-16-13619-2016.

---

## Author Comment (AC1) · 29 Jan 2017

**Review for "Spatial and temporal CCN variations in convection-permitting aerosol microphysics simulations in an idealised marine tropical domain"**

Paper summary and recommendation:

This paper disentangles the contribution of different processes to the overall CCN variability detected over a domain the size of a conventional general circulation model (GCM) grid box for the case of a convective tropical marine boundary layer. The study is performed in a simplified idealised setup. Feedback pathways between aerosol concentrations and the environment via radiative or cloud microphyiscal interactions are ignored. Thereby, an attribution of different processes to CCN variability due to spatial and temporal variability of size and number of the 3 mixed-modes contributing to the CCN budget (Aitken, accumulation and coarse mode) is obtained. The authors show that CCN concentrations may vary up to a factor 3-8 throughout the simulation domain. Understanding the origins of this variability is an important step towards estimating the potential biases of aerosol-cloud interaction estimates obtained by GCMs, which do not resolve this variability. I therefore recommend this article for publication in Atmos. Chem. Phys. following minor revisions.

*We thank the reviewer for their review and note they recognise the paper's value in providing information to explore potential biases in aerosol-cloud interaction estimates from lower spatial resolution GCMs.*

*Our replies to each of the reviewers' comments are provided below (coloured red) and, where changes to the manuscript have been made, these are highlighted in the track-changes version of the document provided.*

Minor Comments – general:

• I believe that your aerosol concentrations are spun up from an entirely clean (i.e. $N_{aero}$=0.0 cm-3) atmosphere. Please state this explicitly in the manuscript. I agree with the authors that this gives you the opportunity to disentangle the individual processes. However, this may be at least partially responsible for the high variability in CCN (800%) obtained after 12h of simulation following the period of intensive updrafts. If that is the case, context should be provided for the interpretation of this estimate. If you initialised a homogeneous profile of e.g. accumulation mode aerosol, would you still obtain such a high degree of variability of CCN following the intense updraft period? Please comment.

*Yes, that's correct – the aerosol concentrations were initialized to zero at the start of the simulation. In the revised manuscript, we have added a sentence to state this explicitly (lines 12-13, page 5). We also agree that the period of intensive updrafts at around 4-7h of simulation is a likely causing an unusually high degree of variability. We do already note the unusual nature of this period in the Abstract (page 1, line 17) and have added "with intense wind-speed conditions" to further suggest the connection between those conditions and increased sea-spray emission.*

• The phrase "strongly convective" period (or conditions) seems to refer to different things throughout the manuscript. Sometimes the phrase seems to be used to refer to the time period of intensive updrafts and strong horizontal winds and sometimes to periods of intense rain fall. Please define this term and use it consistently throughout the manuscript.

*We have clarified use of this phrase in the manuscript (page 1, line 30; page 5, lines 30-31; page 7, lines 5-6; page 8, lines 10-11). Mostly, we refer to the "strongly convective period" as meaning the period when the dynamical conditions (updrafts and horizontal wind speeds) are intense.*

• It has been shown (e.g. Textor et al, 2006: "Analysis and quantification of the diversities of aerosol life cycles within AeroCom") that different assumptions made in modeling the sea salt flux may yield vastly different estimates of sea salt emission fluxes. How sensitive do the authors think their results are to their implemented SS emission parameterisation? Please comment.

*The simulations apply the Gong et al. (2003) sea-spray source function, which includes the behavior of the Monahan et al. (1986) parameterisation, with additional parameter to control emission of ultra-fine sea spray particles. The parameterisation was used by many of the global models in phase 1 of the AeroCom intercomparison, as analysed by Textor et al. (2006). As we also explain in our responses to the other reviewer, in Figure 5 of the recent paper by Salter et al., (2015), several different sea-spray source functions are presented in terms of their emission flux against wind speed, with the Gong being in the mid-range of the different parameterisations. We therefore believe our results are not sensitive to the particular sea-spray emission parameterisations and would be robust if a different emissions scheme were used.*

*Gong, S. L.: A parameterization of sea-salt aerosol source function for sub- and super-micron particles, Global Biogeochem. Cycles., 14, 1097-1103, 2003.*

*Monahan, E. C., Spiel, D. E., and Davidson, K. L.: A model of marine aerosol generation via whitecaps and wave disruption. Oceanic Whitecaps. Edited by EC Monahan and G MacNiochaill, pp 167-193, D Reidel, Norwell, Mass, 1986.*

*Salter, M. E., Zieger, P., Acosta Navarro, J. C., Grythe, H., Kirkevåg, A., Rosati, B., Riipinen, I., and Nilsson, E. D.: An empirically derived inorganic sea spray source function incorporating sea surface temperature, Atmos. Chem. Phys., 15, 11047-11066, doi:10.5194/acp-15-11047-2015, 2015.*

*Textor, C., Schulz, M., Guibert, S., Kinne, S., Balkanski, Y., Bauer, S., Berntsen, T., Berglen, T., Boucher, O., Chin, M., Dentener, F., Diehl, T., Easter, R., Feichter, H., Fillmore, D., Ghan, S., Ginoux, P., Gong, S., Grini, A., Hendricks, J., Horowitz, L., Huang, P., Isaksen, I., Iversen, I., Kloster, S., Koch, D., Kirkevåg, A., Kristjansson, J. E., Krol, M., Lauer, A., Lamarque, J. F., Liu, X., Montanaro, V., Myhre, G., Penner, J., Pitari, G., Reddy, S., Seland, Ø., Stier, P., Takemura, T., and Tie, X.: Analysis and quantification of the diversities of aerosol life cycles within AeroCom, Atmos. Chem. Phys., 6, 1777-1813, doi:10.5194/acp-6-1777-2006, 2006.*

Minor Comments – specific:

• P3L10: Please include reference Zubler et al (2011):"Simulation of dimming and brightening in Europe from 1958 to 2001 using a regional climate model", JGR, doi:10.1029/2010JD015396.
*Done*

• P3L10-L12: Two recent studies have investigated the impact of resolution on aerosol variability and aerosol-cloud interactions in regional climate models down to the kilometre scale for boundary layer clouds. These references should be added:
  ◦ Possner et al (2016): "The resolution dependence of cloud effects and ship-induced aerosol cloud interactions in marine stratocumulus", JGR, doi:10.1002/2015JD024685.
  ◦ Weigum et al (2016): "Effect of aerosol subgrid variability on aerosol optical depth and cloud condensation nuclei: implications for global aerosol modelling", ACP, doi:10.5194/acp-16-13619-2016.
*Thanks -- we added citations to these references in the manuscript.*

• P3L18: Please clarify complexity of aerosol treatment here, as there have been numerous studies investigating the sensitivity of marine deep and shallow convection to simplified aerosol treatments.

*We are not sure what the reviewer refers to here. In the final paragraph of this section we do refer to the simulations applying an aerosol microphysics, and the particular type of aerosol dynamics scheme is clearly described in section 2.1, which follows immediately from this section. We therefore feel the level of detail given in this Introduction part of the manuscript is sufficient.*

• P3L23: Please rephrase "to well characterize".

*By "well characterize" we mean that the model will represent the dominant sources of CCN variability and therefore simulated CCN variations would be expected to be realistic. We therefore feel the word "well" is appropriate here. However, on reflection, perhaps that word does not need to be stated explicitly. We deleted it and also removed the 2ⁿᵈ instance of "influences" later in the sentence to improve the wording.*

• P3L25: How do the authors determine the "realistic" level of variability? Please add references here.
*We cite Yang et al. (2011) as a study that includes a similar level of model complexity.*

*Yang, Q., W. I. Gustafson Jr., Fast, J. D., Wang, H., Easter, R. C., Morrison, H., Lee, Y.-N., Chapman, E. G., Spak, S. N., and Mena-Carrasco, M. A.: Assessing regional scale predictions of aerosols, marine stratocumulus, and their interactions during VOCALS-REx using WRF-Chem, Atmos. Chem. Phys., 11, 11951-11975, doi:10.5194/acp-11-11951-2011, 2011.*

• P4L30: Please rephrase "only a short demonstration simulation is here carried out".
*Done*

• P4L31: Please rephrase "carried on".
*Done*

• P5L11: Please rephrase "becomes precipitating", "becoming more intense".
*Done*

• P5L22: Please rephrase "associated cold pooling".
*Done*

• P6L24: By which criterion do you define your simulation to have fully spun up? Please clarify.
*As discussed earlier in the manuscript, the model is spinning up over the simulation, including the period of convective instability at ~6h. Whilst we do not have specific criteria, we consider our focus on the last 12 hours of the simulation to be after the initial spin-up of the dynamics and primary aerosol (sea spray) in the model. We acknowledge that the secondary sulphate particles may still be spinning up, and we do discuss this clearly within the existing manuscript text.*

• P7L20: The correspondence between patterns in highest particle concentrations and smaller particle sizes in Fig. 3 is not obvious to me in this particular figure. Please elaborate, or remove comment.
*As this is not a key element of the study we decided to remove this comment.*

• P7L31ff: The second half of the day is not only characterised by calmer wind conditions, but also by intense precipitation between 12 – 18 h. I believe that it should be mentioned here.
*Yes – we agree. We now refer to this at this point as the reviewer suggest.*

• P8L31: "adjusts to the very strong sea-salt emission and wet removal". However precipitation only really intensifies much later than 8h after initialisation. Please comment on the role of wet removal during this period.

*Even if the precipitation rate is less intense than later in the simulation, it is in average equal to 10 mm/h over the 5-8h period. So, the wet removal process becomes effective at this period of precipitation onset. We clarified this in the manuscript.*

• P9L31 – P10L1: Remove sentence "The relative decrease in …". You already stated that it is linear. *Done*

• P10L30: "… sea-salt aerosol are transported vertically by turbulent diffusion", I would have thought that the convective updrafts would also contribute? Please comment, or adapt text.
*You are right. We clarified the manuscript.*

• P11L9: Please rephrase "wind speeds condition".
*Done*

• P11L21ff: The authors state that CCN variations can be as large as factor 8. This number is obtained 12h after the simulation (Fig. 8). At this time the winds subside and precipitation builds up. So, how well does it characterise the CCN variability obtained during the period of intense updrafts? It may be helpful to include a box diagram for 6h after initialisation in Fig. 8.
*We explain already that the processes mean we consider the CCN variability to be well characterized in the simulations. As above, this strongly convectively unstable period is not so representative of typical conditions and we therefore feel additional box diagram is not needed.*

Furthermore the authors state that the CCN variability is large whilst the accumulation mode variability is smaller. This is confusing as I would assume most CCN to stem from the accumulation mode (see Fig 7.). Please clarify.

• Fig 1: For illustrative purposes the authors may consider adding a line of adiabatic parcel ascent.
*We added a line representing the adiabatic parcel ascent and its specific levels (LCL, CCL and LFC).*

• Fig 2: Please rephrase "mean total top cloud height" to "mean cloud top height".
*Done.*

Please rephrase "rain accumulation" to "accumulated rain" or "accumulated precipitation".
*Done.*

• Fig 4: For clarification it may help adding day and night markers for the sulfate chemistry.
*As described in the section 2, there is no diurnal cycle in the model (page 4, line 30).*

• Fig6 and Fig7: Does your aerosol scheme specify modal boundaries for the Aitken, accumulation and coarse mode? If so what are these? These could be added in the model description section.
*The process of mode-merging is explained in Mann et al. (2010), and yes the scheme includes so-called "separation diameters" which determine at what point the particle size has grown large enough to be transferred to the adjacent larger mode. The values used are those as revised in Mann et al. (2012) to better capture size distributions simulated with the more complex sectional aerosol scheme. We feel the existing references are adequate here and think it is not necessary to re-state the values used explicitly.*

*Mann, G. W., Carslaw, K. S., Spracklen, D. V., Ridley, D. A., Manktelow, P. T., Chipperfield, M. P., Pickering, S. J., and Johnson, C. E.: Description and evaluation of GLOMAP-mode: a modal global aerosol microphysics model for the UKCA composition-climate model, Geosci. Model Dev., 3, 519-551, 2010.*

*Mann, G. W., Carslaw, K. S., Ridley, D. A., Spracklen, D. V., Pringle, K. J., Merikanto, J., Korhonen, H., Schwarz, J. P., Lee, L. A., Manktelow, P. T., Woodhouse, M. T., Schmidt, A., Breider, T. J., Emmerson, K. M., Reddington, C. L., Chipperfield, M. P., and Pickering, S. J.: Inter-comparison of modal and sectional aerosol microphysics representations within the same 3-D global chemical transport model, Atmos. Chem. Phys., 12, 4449-4476, 2012.*

• Fig 7: What causes the large variability in radius for the accumulation mode particles up to 8h after initialisation? This is discussed in the text on P8L30ff, but I would have thought that the SS emission radius would be tighter constrained and that wet removal processes play a larger role later during the simulation (after 12h) as the RR peaks. Please clarify.

*The large variability in radius of accumulation mode particles is caused by several processes, including those mentioned by the reviewer. The model size distribution responds to the different rapid changes during this high wind speed period that is generating strong sea-spray emissions. We therefore expect both emissions effects and removal effects to be influencing the behavior of the model. The different influences are complex and we feel our current qualitative discussion in the text is sufficient.*

---

## Author Comment (AC2) · 29 Jan 2017

**Review for: "Spatial and temporal CCN variations in convection permitting aerosol microphysics simulations in an idealised marine tropical domain"**

**Summary and recommendation:**

This article employs a convection-permitting resolution model to assess the contribution of spatial and temporal variations in aerosol properties for the case of a convective tropical marine boundary layer to CCN variability across a domain the size of a GCM grid box. The model is setup in a simplified idealised configuration in which the radiation scheme was turned off and CCN concentrations do not feed through to the cloud microphysics. Subsequently, the current setup ignores feedbacks associated with aerosol-radiative and aerosol-cloud microphysical interactions that may impact the simulated aerosol field in the model. The authors find that the simulated CCN concentrations can vary significantly over the domain, more than a factor of 8 during strongly convective conditions. They assess the contribution of dynamical, chemical and microphysical processes to this high variability in CCN and attribute it to increased sea salt/DMS emissions when spatial and temporal wind speed fluctuations become resolved at this convection-permitting resolution, increasing peak wind-speeds. This is an interesting finding as current GCMs cannot explicitly resolve sub-grid scale variability in wind speeds. Such modelling frameworks are required to elicit the impact of spatial/temporal resolution in GCMs on the representation of aerosol-cloud interactions. Therefore, I recommend publication of this article ACP once the following revisions have been addressed:

*We thank the reviewer for their constructive review and address each of the specific comments raised below with our responses coloured red, and, where changes to the manuscript have been made, these are highlighted in the track-changes version of the document provided.*

**General comments:**

- The modelling framework developed is described as ground-breaking. One of the key advantages of the model is in the use of a unified modelling (UM) framework to investigate the dependence of parameters involved in aerosol-cloud interactions on model resolution. However, this strength has not been captivated upon in this study. There is a lack of evaluation of the impact of the increased resolution in the model on the parameters of interest. A comparison of the domain averaged parameter values presented in the study to the same parameters simulated by the GCM would be highly beneficial and greatly strengthen the conclusions presented. Does the observed sub-grid scale variability in CCN impact the average CCN concentration across the domain compared to a GCM? This comparison should be provided before publication in ACP.

*In our Figures 8 and 9 we specifically assess the CCN variability in the model, presenting the full pdf of the distribution in the vertical and at different points of the simulation across varying wind speed conditions. We agree strongly that comparing this CCN variability against that found from a lower resolution simulation with the same GCM would be a valuable comparison to make. Indeed the capability to nest down from global simulations with parameterized convection to limited-area domains runs at convection permitting resolution will enable this to be quantified in future studies with this Unified Model (UM) framework. However, the simulations here are with an idealized configuration of the UM whereby the solar variations*

*across a daily cycle are de-activated. We therefore feel comparing CCN variability across different horizontal resolution would best be reserved for a future study with the nested UM.*

- CCN represent the aerosol particles that can form cloud droplets under reasonable atmospheric supersaturations. Accordingly, CCN concentrations always refer to a specific supersaturation, for example, CCN (0.1%) or CCN (0.5%) and one should be careful when comparing CCN concentrations measured or simulated at different supersaturations. What supersaturation was used throughout the article for the CCN concentrations presented?

*Yes, we should have specified that, as in Mann et al. (2012), CCN concentrations are calculated here as soluble particles with dry diameter larger than 50nm, which corresponds to a supersaturation of 0.35%, calculated by Kohler theory for a pure sulphuric acid solution droplet. The marine aerosol review article by O'Dowd et al. (1997) refers to this threshold size as a good representative for activating nuclei. The revised manuscript now gives this definition for CCN at first use in the results section, referring to the specific threshold size and supersaturation used in the calculations.*

The variability in CCN concentrations reported in this idealised configuration has been shown to be strongly dependant on variability in wind speed across the domain. This is unsurprising considering the strong wind-speed dependence of the sea spray emission parameterisation employed. Accordingly more discussion is required as to the sensitivity of the results presented on the choice of sea spray emission parameterisation with regard to the following:

- As the findings presented are strongly linked to the simulated wind speed field across the domain some discussion is required as to how accurate the simulated wind field and convective perturbation is compared to the real world. Also, is the aerosol, thus, CCN variability simulated expected compared to observations? Please discuss in relation to the footprint of flux measurements performed to measure sea-salt emissions in the marine environment and associated variability observed from these measurement campaigns.

*We consider the model wind speed field across the domain, as simulated by the atmospheric dynamics in the MetUM, to be, for this type of model, highly realistic since the convection is explicitly resolved (no convection parameterization is required). The reviewer refers to the temporal footprint of flux measurements used to measure sea-salt emission. As we discuss in the reply to their later comment, that's an interesting point in relation to the time-window for the flux measurements used in deriving and evaluating the sea-spray source function flux used in our study. We consider this to be part of the future analysis in terms of applying the nested UM framework to assess the CCN variability in simulations with different spatial resolution.*

- Numerous sea-salt emission parameterisations exist, derived from a variety of in-situ measurement campaigns and laboratory experiments. How does the chosen parameterisations wind-speed dependence compare to the range of parameterisations in the literature, e.g. Fig. 5 Salter et al., 2015? How might a different parameterisation alter the high variability in CCN across the domain found?

*As we explain in the text, sea-salt is emitted according to the Gong (2003) sea-spray emission parameterization, as applied in our global model simulations (e.g. Mann et al., 2010). The Gong parameterization is based on the Monahan et al. (1986) sea-spray source function with a parameterization to better capture emissions of ultra-fine sea-spray. We have set the theta parameter controlling these ultra-fine sea-sprays to 30 in these simulations, as applied in Gong (2003). As in our reply to Reviewer 1, the Gong-Monahan emissions flux lie within the mid-range of other parameterizations in the literature (see e.g. Figure 5 of Salter et al., 2015),*

*so, in our view, it is reasonable to expect our results to be robust, with this sea-spray emissions parameterization particularly designed to capture the emitted ultra-fine sea spray.*

• The onset of wave breaking is important for sea spray aerosol formation. It is generally recognized that the whitecap fraction and therefore sea spray aerosol production is zero for wind speeds less than ~ 3 m s$^{-1}$ (Blanchard, 1963; Monahan, 1971). The implication of this with respect to the findings requires discussion, for example, what is the contribution of the total CCN variability simulated between 0-3 m s$^{-1}$? At what wind speeds does the CCN concentration begin to increase sharply, is there a threshold value?

*There is no threshold velocity in the sea-spray source function, but, since the Gong parameterization is based on Monahan et al. (1986), it applies the emissions flux to be proportional to the 10m wind speed to the power 3.41, so the increase is quite steep as wind speed increases. Similarly, at the low windspeeds range mentioned (0-3 m/s) emissions fluxes will be low, and the non-linear dependence of the sea-spray emissions is one reason why this additional CCN variability becomes higher at this higher spatial resolution. We point out in the Introduction (page 3 lines 6-8) and feel this is explored sufficiently with the current text.*

• Discussion is required on the applicability of the chosen parameterisation of the resolution of the model (1Km) and time-step. Typically sea salt emission parameterisations are applicable to certain footprints, and parameterisations developed from in-situ observations are dependent on the memory of the wave field (a rising sea will result in a different emission profile than a falling sea). In addition parameterisations are developed using longer time windows for averaging for flux measurements compared to the model time-step employed. Is the sea spray source function being applied in the model at this temporal/spatial resolution in the way it was designed?

*As we state in the manuscript (page 5, lines 4-5), emissions are calculated (and enacted) every timestep of the simulation, which is 30 seconds at this high spatial resolution. Although the reviewer is absolutely correct to point out that wave state of the sea surface affects emitted sea-spray (e.g. Grythe et al., 2014, ACP), in our simulations with the Gong (2003) parameterization, these affects are not included. To address the reviewer comments we added the following to the revised manuscript:*
*"Other influences such as changes in sea surface wave state will also influence sea spray emissions (e.g. Grythe et al., 2014), but these effects are not resolved in this study. The Gong-Monahan parameterization used here is based on sea spray flux measurements made over a longer time period than the model timestep (30s), and observating capabilities now include eddy covariance sea-spray flux measurements (e.g. Norris et al., 2012), we expect our approach will resolve the dominant sources of sea spray emissions flux variability."*

**Minor comments:**

• Section 2: A figure of the modified sea-spray source function used in the study would be beneficial here, especially for experimentalists.

*We feel it is sufficient to reference the Gong (2003) paper. It's an established parameterization and was recommended for models to use in the AeroCom phase 1 co-ordinated experiment.*

• Section 2: For a modelling framework described as ground-breaking the model description is relatively sparse, for instance, how is hygroscopic growth parameterised in the model? This

will affect the evolution of the aerosol field across the domain. Please provide a more detailed description of the aerosol microphysics scheme.

*The ground-breaking aspect of this study is the ability to use a new numerical framework that is based on a coupling between the UKCA detailed aerosol module and the MetUM model at very high scale.*

*Regarding the hygroscopic growth, this aerosol process is parameterized thanks to the ZSR method (Zadanovksii, 1948; Stokes and Robinson, 1966) using data from Jacobson et al. (1996) to calculate the binary electrolyte molalities. The complete description of the hygroscopic growth parameterization as well as the description of all the others aerosol processes of the model are described in details in Mann et al. (2010).*

*We modified the manuscript in order to inform the reader that a complete description of the different aerosol processes is provided in this paper.*

• Section 2.2: It is widely known 1-moment cloud microphysics schemes introduce errors compared to 2 or3 moment schemes. Some justification of this choice is required, was it due to computational restraints?

*Yes, in this study a single-moment microphysics scheme is used as it is the only one currently available in the latest version of the Unified Model at that time. We clarified the description of the microphysics scheme in the manuscript.*

• Section 3.2, line 8: "Aitken mode are almost exclusively secondary in nature": Please reword, this is two strong, studies exist which show emission of sea spray in this size regime, e.g. Salter et al., 2015.

*We clarified the manuscript.*

• Section 4, line 25. "comprising two elements": reword.

*We clarified the manuscript.*

• Fig.4: Why do the error bars in DMS & SO2/H2SO4 not correspond? Some discussion on expected oxidation timescales required, why is there no offset between H2SO4 & DMS observed?

*We explain in the text that the steps involved for the SO2 and H2SO4 to be produced following oxidation in the atmosphere and we do not understand the reviewer's point here. We feel the existing text is sufficient here to explain what is shown in the Figure.*

• Recent studies have probed the dependence of aerosol processes on model resolution, for instance Weigum et al., 2016. This should be referenced.

*We added this reference in the introduction section.*

**References:**

Blanchard, D. C.: The electrification of the atmosphere by particles from bubbles in the sea, Prog. Oceanogr., 1, 171–202, 1963.

Monahan, E. C.: Oceanic whitecaps, J. Phys. Oceanogr., 1, 139– 144, 1971.

Salter, M. E., Zieger, P., Acosta Navarro, J. C., Grythe, H., Kirkevåg, A., Rosati, B., Riipinen, I., and Nilsson, E. D.: An empirically derived inorganic sea spray source function incorporating sea surface temperature, Atmos. Chem. Phys., 15, 11047-11066, doi:10.5194/acp-15-11047-2015, 2015.

Weigum et al.: Effect of aerosol subgrid variability on aerosol optical depth and cloud condensation nuclei: implications for global aerosol modelling, ACP doi:10.5194/acp-16-13619-2016.

*Gong, S. L.: A parameterization of sea-salt aerosol source function for sub- and super-micron particles, Global Biogeochem. Cycles., 14, 1097-1103, 2003.*

*Grythe H., Ström, J., Krejci, R., Quinn, P. and Stohl, A., A review of sea-spray aerosol source functions using a large global set of sea salt aerosol concentration measurements, Atmos. Chem. Phys., 14, 1277–1297, 2014.*

*Jacobson, M. Z., Tabazadeh, A., and Turco, R. P.: Simulating equilibrium within aerosols and non-equilibrium between gases and aerosols, J. Geophys. Res., 101(D4), 9079–9091, 1996.*

*Mann, G. W., Carslaw, K. S., Spracklen, D. V., Ridley, D. A., Manktelow, P. T., Chipperfield, M. P., Pickering, S. J., and Johnson, C. E.: Description and evaluation of GLOMAP-mode: a modal global aerosol microphysics model for the UKCA composition-climate model, Geosci. Model Dev., 3, 519-551, 2010.*

*Mann, G. W., Carslaw, K. S., Ridley, D. A., Spracklen, D. V., Pringle, K. J., Merikanto, J., Korhonen, H., Schwarz, J. P., Lee, L. A., Manktelow, P. T., Woodhouse, M. T., Schmidt, A., Breider, T. J., Emmerson, K. M., Reddington, C. L., Chipperfield, M. P., and Pickering, S. J.: Intercomparison of modal and sectional aerosol microphysics representations within the same 3-D global chemical transport model, Atmos. Chem. Phys., 12, 4449-4476, doi:10.5194/acp-12-4449-2012, 2012.*

*Monahan, E. C., Spiel, D. E., and Davidson, K. L.: A model of marine aerosol generation via whitecaps and wave disruption. Oceanic Whitecaps. Edited by EC Monahan and G MacNiochaill, pp 167-193, D Reidel, Norwell, Mass, 1986.*

*Norris, S. J., Brooks, I. M., Hill, M. K. Brooks, B. J., Smith, M. H., Sproson, D. A. J.: Eddy covariance measurements of the sea spray aerosol flux over the open ocean, J. Geophys. Res., vol. 117, doi:10.1029/2011JD016549, 2012.*

*O'Dowd, C. D., Smith, M. H., Consterdine, I. E., and Lowe, J. A.: Marine aerosol, sea salt, and the marine sulphur cycle: a short review, Atmos. Environ., 31, 73-80, 1997.*

*Salter, M. E., Zieger, P., Acosta Navarro, J. C., Grythe, H., Kirkevåg, A., Rosati, B., Riipinen, I., and Nilsson, E. D.: An empirically derived inorganic sea spray source function incorporating sea surface temperature, Atmos. Chem. Phys., 15, 11047-11066, doi:10.5194/acp-15-11047-2015, 2015.*

*Stokes, R. H. and Robinson, R. A.: Interactions in aqueous non-electrolyte solutions. I. Solute-solvent equilibria, J. Phys. Chem., 70, 2126–2130, 1966.*

*Zadanovskii, A. B.: New methods for calculating solubilities of electrolytes in multicomponent systems, Zh. Fiz. Khim., 22, 1475–1485, 1948.*

---

## Editor Decision (ED1)

**Review for "Spatial and temporal CCN variations in convection-permitting aerosol microphysics simulations in an idealised marine tropical domain"**

Paper summary and recommendation:

This paper disentangles the contribution of different processes to the overall CCN variability detected over a domain the size of a conventional general circulation model (GCM) grid box for the case of a convective tropical marine boundary layer. The study is performed in a simplified idealised setup. Feedback pathways between aerosol concentrations and the environment via radiative or cloud microphyiscal interactions are ignored. Thereby, an attribution of different processes to CCN variability due to spatial and temporal variability of size and number of the 3 mixed-modes contributing to the CCN budget (Aitken, accumulation and coarse mode) is obtained. The authors show that CCN concentrations may vary up to a factor 3-8 throughout the simulation domain. Understanding the origins of this variability is an important step towards estimating the potential biases of aerosol-cloud interaction estimates obtained by GCMs, which do not resolve this variability. I therefore recommend this article for publication in Atmos. Chem. Phys. following minor revisions.

*We thank the reviewer for their review and note they recognise the paper's value in providing information to explore potential biases in aerosol-cloud interaction estimates from lower spatial resolution GCMs.*

*Our replies to each of the reviewers' comments are provided below (coloured red) and, where changes to the manuscript have been made, these are highlighted in the track-changes version of the document provided.*

Minor Comments – general:

• I believe that your aerosol concentrations are spun up from an entirely clean (i.e. Naero=0.0 cm-3) atmosphere. Please state this explicitly in the manuscript. I agree with the authors that this gives you the opportunity to disentangle the individual processes. However, this may be at least partially responsible for the high variability in CCN (800%) obtained after 12h of simulation
 following the period of intensive updrafts. If that is the case, context should be provided for the interpretation of this estimate. If you initialised a homogeneous profile of e.g. accumulation mode aerosol, would you still obtain such a high degree of variability of CCN following the intense updraft period? Please comment.

*Yes, that's correct – the aerosol concentrations were initialized to zero at the start of the simulation. In the revised manuscript, we have added a sentence to state this explicitly (lines 12-13, page 5). We also agree that the period of intensive updrafts at around 4-7h of simulation is a likely causing an unusually high degree of variability. We do already note the unusual nature of this period in the Abstract (page 1, line 17) and have added "with intense wind-speed conditions" to further suggest the connection between those conditions and increased sea-spray emission.*

• The phrase "strongly convective" period (or conditions) seems to refer to different things throughout the manuscript. Sometimes the phrase seems to be used to refer to the time period of intensive updrafts and strong horizontal winds and sometimes to periods of intense rain fall. Please define this term and use it consistently throughout the manuscript.

*We have clarified use of this phrase in the manuscript (page 1, line 30; page 5, lines 30-31; page 7, lines 5-6; page 8, lines 10-11). Mostly, we refer to the "strongly convective period" as meaning the period when the dynamical conditions (updrafts and horizontal wind speeds) are intense.*

• It has been shown (e.g. Textor et al, 2006: "Analysis and quantification of the diversities of aerosol life cycles within AeroCom") that different assumptions made in modeling the sea salt flux may yield vastly different estimates of sea salt emission fluxes. How sensitive do the authors think their results are to their implemented SS emission parameterisation? Please comment.

*The simulations apply the Gong et al. (2003) sea-spray source function, which includes the behavior of the Monahan et al. (1986) parameterisation, with additional parameter to control emission of ultra-fine sea spray particles. The parameterisation was used by many of the global models in phase 1 of the AeroCom intercomparison, as analysed by Textor et al. (2006). As we also explain in our responses to the other reviewer, in Figure 5 of the recent paper by Salter et al., (2015), several different sea-spray source functions are presented in terms of their emission flux against wind speed, with the Gong being in the mid-range of the different parameterisations. We therefore believe our results are not sensitive to the particular sea-spray emission parameterisations and would be robust if a different emissions scheme were used.*

*Gong, S. L.: A parameterization of sea-salt aerosol source function for sub- and super-micron particles, Global Biogeochem. Cycles., 14, 1097-1103, 2003.*

*Monahan, E. C., Spiel, D. E., and Davidson, K. L.: A model of marine aerosol generation via whitecaps and wave disruption. Oceanic Whitecaps. Edited by EC Monahan and G MacNiochaill, pp 167-193, D Reidel, Norwell, Mass, 1986.*

*Salter, M. E., Zieger, P., Acosta Navarro, J. C., Grythe, H., Kirkevåg, A., Rosati, B., Riipinen, I., and Nilsson, E. D.: An empirically derived inorganic sea spray source function incorporating sea surface temperature, Atmos. Chem. Phys., 15, 11047-11066, doi:10.5194/acp-15-11047-2015, 2015.*

*Textor, C., Schulz, M., Guibert, S., Kinne, S., Balkanski, Y., Bauer, S., Berntsen, T., Berglen, T., Boucher, O., Chin, M., Dentener, F., Diehl, T., Easter, R., Feichter, H., Fillmore, D., Ghan, S., Ginoux, P., Gong, S., Grini, A., Hendricks, J., Horowitz, L., Huang, P., Isaksen, I., Iversen, I., Kloster, S., Koch, D., Kirkevåg, A., Kristjansson, J. E., Krol, M., Lauer, A., Lamarque, J. F., Liu, X., Montanaro, V., Myhre, G., Penner, J., Pitari, G., Reddy, S., Seland, Ø., Stier, P., Takemura, T., and Tie, X.: Analysis and quantification of the diversities of aerosol life cycles within AeroCom, Atmos. Chem. Phys., 6, 1777-1813, doi:10.5194/acp-6-1777-2006, 2006.*

Minor Comments – specific:

• P3L10: Please include reference Zubler et al (2011):"Simulation of dimming and brightening in Europe from 1958 to 2001 using a regional climate model", JGR, doi:10.1029/2010JD015396.
*Done*

• P3L10-L12: Two recent studies have investigated the impact of resolution on aerosol variability and aerosol-cloud interactions in regional climate models down to the kilometre scale for boundary layer clouds. These references should be added:
 ◦ Possner et al (2016): "The resolution dependence of cloud effects and ship-induced aerosol cloud interactions in marine stratocumulus", JGR, doi:10.1002/2015JD024685.
 ◦ Weigum et al (2016): "Effect of aerosol subgrid variability on aerosol optical depth and cloud condensation nuclei: implications for global aerosol modelling", ACP, doi:10.5194/acp-16-13619-2016.
*Thanks -- we added citations to these references in the manuscript.*

• P3L18: Please clarify complexity of aerosol treatment here, as there have been numerous studies investigating the sensitivity of marine deep and shallow convection to simplified aerosol treatments.

*We are not sure what the reviewer refers to here. In the final paragraph of this section we do refer to the simulations applying an aerosol microphysics, and the particular type of aerosol dynamics scheme is clearly described in section 2.1, which follows immediately from this section. We therefore feel the level of detail given in this Introduction part of the manuscript is sufficient.*

• P3L23: Please rephrase "to well characterize".

*By "well characterize" we mean that the model will represent the dominant sources of CCN variability and therefore simulated CCN variations would be expected to be realistic. We therefore feel the word "well" is appropriate here. However, on reflection, perhaps that word does not need to be stated explicitly. We deleted it and also removed the 2nd instance of "influences" later in the sentence to improve the wording.*

• P3L25: How do the authors determine the "realistic" level of variability? Please add references here.
*We cite Yang et al. (2011) as a study that includes a similar level of model complexity.*

*Yang, Q., W. I. Gustafson Jr., Fast, J. D., Wang, H., Easter, R. C., Morrison, H., Lee, Y.-N., Chapman, E. G., Spak, S. N., and Mena-Carrasco, M. A.: Assessing regional scale predictions of aerosols, marine stratocumulus, and their interactions during VOCALS-REx using WRF-Chem, Atmos. Chem. Phys., 11, 11951-11975, doi:10.5194/acp-11-11951-2011, 2011.*

• P4L30: Please rephrase "only a short demonstration simulation is here carried out".
*Done*

• P4L31: Please rephrase "carried on".
*Done*

• P5L11: Please rephrase "becomes precipitating", "becoming more intense".
*Done*

• P5L22: Please rephrase "associated cold pooling".
*Done*

• P6L24: By which criterion do you define your simulation to have fully spun up? Please clarify.
*As discussed earlier in the manuscript, the model is spinning up over the simulation, including the period of convective instability at ~6h. Whilst we do not have specific criteria, we consider our focus on the last 12 hours of the simulation to be after the initial spin-up of the dynamics and primary aerosol (sea spray) in the model. We acknowledge that the secondary sulphate particles may still be spinning up, and we do discuss this clearly within the existing manuscript text.*

• P7L20: The correspondence between patterns in highest particle concentrations and smaller particle sizes in Fig. 3 is not obvious to me in this particular figure. Please elaborate, or remove comment.
*As this is not a key element of the study we decided to remove this comment.*

• P7L31ff: The second half of the day is not only characterised by calmer wind conditions, but also by intense precipitation between 12 – 18 h. I believe that it should be mentioned here.
*Yes – we agree. We now refer to this at this point as the reviewer suggest.*

• P8L31: "adjusts to the very strong sea-salt emission and wet removal". However precipitation only really intensifies much later than 8h after initialisation. Please comment on the role of wet removal during this period.

*Even if the precipitation rate is less intense than later in the simulation, it is in average equal to 10 mm/h over the 5-8h period. So, the wet removal process becomes effective at this period of precipitation onset. We clarified this in the manuscript.*

• P9L31 – P10L1: Remove sentence "The relative decrease in …". You already stated that it is linear. *Done*

• P10L30: "… sea-salt aerosol are transported vertically by turbulent diffusion", I would have thought that the convective updrafts would also contribute? Please comment, or adapt text.
*You are right. We clarified the manuscript.*

• P11L9: Please rephrase "wind speeds condition".
*Done*

• P11L21ff: The authors state that CCN variations can be as large as factor 8. This number is obtained 12h after the simulation (Fig. 8). At this time the winds subside and precipitation builds up. So, how well does it characterise the CCN variability obtained during the period of intense updrafts? It may be helpful to include a box diagram for 6h after initialisation in Fig. 8.
*We explain already that the processes mean we consider the CCN variability to be well characterized in the simulations. As above, this strongly convectively unstable period is not so representative of typical conditions and we therefore feel additional box diagram is not needed.*

Furthermore the authors state that the CCN variability is large whilst the accumulation mode variability is smaller. This is confusing as I would assume most CCN to stem from the accumulation mode (see Fig 7.). Please clarify.

• Fig 1: For illustrative purposes the authors may consider adding a line of adiabatic parcel ascent.
*We added a line representing the adiabatic parcel ascent and its specific levels (LCL, CCL and LFC).*

• Fig 2: Please rephrase "mean total top cloud height" to "mean cloud top height".
*Done.*

Please rephrase "rain accumulation" to "accumulated rain" or "accumulated precipitation".
*Done.*

• Fig 4: For clarification it may help adding day and night markers for the sulfate chemistry.
*As described in the section 2, there is no diurnal cycle in the model (page 4, line 30).*

• Fig6 and Fig7: Does your aerosol scheme specify modal boundaries for the Aitken, accumulation and coarse mode? If so what are these? These could be added in the model description section.
*The process of mode-merging is explained in Mann et al. (2010), and yes the scheme includes so-called "separation diameters" which determine at what point the particle size has grown large enough to be transferred to the adjacent larger mode. The values used are those as revised in Mann et al. (2012) to better capture size distributions simulated with the more complex sectional aerosol scheme. We feel the existing references are adequate here and think it is not necessary to re-state the values used explicitly.*

*Mann, G. W., Carslaw, K. S., Spracklen, D. V., Ridley, D. A., Manktelow, P. T., Chipperfield, M. P., Pickering, S. J., and Johnson, C. E.: Description and evaluation of GLOMAP-mode: a modal global aerosol microphysics model for the UKCA composition-climate model, Geosci. Model Dev., 3, 519-551, 2010.*

*Mann, G. W., Carslaw, K. S., Ridley, D. A., Spracklen, D. V., Pringle, K. J., Merikanto, J., Korhonen, H., Schwarz, J. P., Lee, L. A., Manktelow, P. T., Woodhouse, M. T., Schmidt, A., Breider, T. J., Emmerson, K. M., Reddington, C. L., Chipperfield, M. P., and Pickering, S. J.: Inter-comparison of modal and sectional aerosol microphysics representations within the same 3-D global chemical transport model, Atmos. Chem. Phys., 12, 4449-4476, 2012.*

• Fig 7: What causes the large variability in radius for the accumulation mode particles up to 8h after initialisation? This is discussed in the text on P8L30ff, but I would have thought that the SS emission radius would be tighter constrained and that wet removal processes play a larger role later during the simulation (after 12h) as the RR peaks. Please clarify.

*The large variability in radius of accumulation mode particles is caused by several processes, including those mentioned by the reviewer. The model size distribution responds to the different rapid changes during this high wind speed period that is generating strong sea-spray emissions. We therefore expect both emissions effects and removal effects to be influencing the behavior of the model. The different influences are complex and we feel our current qualitative discussion in the text is sufficient.*

**Review for: "Spatial and temporal CCN variations in convection permitting aerosol microphysics simulations in an idealised marine tropical domain"**

**Summary and recommendation:**

This article employs a convection-permitting resolution model to assess the contribution of spatial and temporal variations in aerosol properties for the case of a convective tropical marine boundary layer to CCN variability across a domain the size of a GCM grid box. The model is setup in a simplified idealised configuration in which the radiation scheme was turned off and CCN concentrations do not feed through to the cloud microphysics. Subsequently, the current setup ignores feedbacks associated with aerosol-radiative and aerosol-cloud microphysical interactions that may impact the simulated aerosol field in the model. The authors find that the simulated CCN concentrations can vary significantly over the domain, more than a factor of 8 during strongly convective conditions. They assess the contribution of dynamical, chemical and microphysical processes to this high variability in CCN and attribute it to increased sea salt/DMS emissions when spatial and temporal wind speed fluctuations become resolved at this convection-permitting resolution, increasing peak wind-speeds. This is an interesting finding as current GCMs cannot explicitly resolve sub-grid scale variability in wind speeds. Such modelling frameworks are required to elicit the impact of spatial/temporal resolution in GCMs on the representation of aerosol-cloud interactions. Therefore, I recommend publication of this article ACP once the following revisions have been addressed:

*We thank the reviewer for their constructive review and address each of the specific comments raised below with our responses coloured red, and, where changes to the manuscript have been made, these are highlighted in the track-changes version of the document provided.*

**General comments:**

- The modelling framework developed is described as ground-breaking. One of the key advantages of the model is in the use of a unified modelling (UM) framework to investigate the dependence of parameters involved in aerosol-cloud interactions on model resolution. However, this strength has not been captivated upon in this study. There is a lack of evaluation of the impact of the increased resolution in the model on the parameters of interest. A comparison of the domain averaged parameter values presented in the study to the same parameters simulated by the GCM would be highly beneficial and greatly strengthen the conclusions presented. Does the observed sub-grid scale variability in CCN impact the average CCN concentration across the domain compared to a GCM? This comparison should be provided before publication in ACP.

*In our Figures 8 and 9 we specifically assess the CCN variability in the model, presenting the full pdf of the distribution in the vertical and at different points of the simulation across varying wind speed conditions. We agree strongly that comparing this CCN variability against that found from a lower resolution simulation with the same GCM would be a valuable comparison to make. Indeed the capability to nest down from global simulations with parameterized convection to limited-area domains runs at convection permitting resolution will enable this to be quantified in future studies with this Unified Model (UM) framework. However, the simulations here are with an idealized configuration of the UM whereby the solar variations*

*across a daily cycle are de-activated. We therefore feel comparing CCN variability across different horizontal resolution would best be reserved for a future study with the nested UM.*

• CCN represent the aerosol particles that can form cloud droplets under reasonable atmospheric supersaturations. Accordingly, CCN concentrations always refer to a specific supersaturation, for example, CCN (0.1%) or CCN (0.5%) and one should be careful when comparing CCN concentrations measured or simulated at different supersaturations. What supersaturation was used throughout the article for the CCN concentrations presented?

*Yes, we should have specified that, as in Mann et al. (2012), CCN concentrations are calculated here as soluble particles with dry diameter larger than 50nm, which corresponds to a supersaturation of 0.35%, calculated by Kohler theory for a pure sulphuric acid solution droplet. The marine aerosol review article by O'Dowd et al. (1997) refers to this threshold size as a good representative for activating nuclei. The revised manuscript now gives this definition for CCN at first use in the results section, referring to the specific threshold size and supersaturation used in the calculations.*

The variability in CCN concentrations reported in this idealised configuration has been shown to be strongly dependant on variability in wind speed across the domain. This is unsurprising considering the strong wind-speed dependence of the sea spray emission parameterisation employed. Accordingly more discussion is required as to the sensitivity of the results presented on the choice of sea spray emission parameterisation with regard to the following:

• As the findings presented are strongly linked to the simulated wind speed field across the domain some discussion is required as to how accurate the simulated wind field and convective perturbation is compared to the real world. Also, is the aerosol, thus, CCN variability simulated expected compared to observations? Please discuss in relation to the footprint of flux measurements performed to measure sea-salt emissions in the marine environment and associated variability observed from these measurement campaigns.

*We consider the model wind speed field across the domain, as simulated by the atmospheric dynamics in the MetUM, to be, for this type of model, highly realistic since the convection is explicitly resolved (no convection parameterization is required). The reviewer refers to the temporal footprint of flux measurements used to measure sea-salt emission. As we discuss in the reply to their later comment, that's an interesting point in relation to the time-window for the flux measurements used in deriving and evaluating the sea-spray source function flux used in our study. We consider this to be part of the future analysis in terms of applying the nested UM framework to assess the CCN variability in simulations with different spatial resolution.*

• Numerous sea-salt emission parameterisations exist, derived from a variety of in-situ measurement campaigns and laboratory experiments. How does the chosen parameterisations wind-speed dependence compare to the range of parameterisations in the literature, e.g. Fig. 5 Salter et al., 2015? How might a different parameterisation alter the high variability in CCN across the domain found?

*As we explain in the text, sea-salt is emitted according to the Gong (2003) sea-spray emission parameterization, as applied in our global model simulations (e.g. Mann et al., 2010). The Gong parameterization is based on the Monahan et al. (1986) sea-spray source function with a parameterization to better capture emissions of ultra-fine sea-spray. We have set the theta parameter controlling these ultra-fine sea-sprays to 30 in these simulations, as applied in Gong (2003). As in our reply to Reviewer 1, the Gong-Monahan emissions flux lie within the mid-range of other parameterizations in the literature (see e.g. Figure 5 of Salter et al., 2015),*

*so, in our view, it is reasonable to expect our results to be robust, with this sea-spray emissions parameterization particularly designed to capture the emitted ultra-fine sea spray.*

• The onset of wave breaking is important for sea spray aerosol formation. It is generally recognized that the whitecap fraction and therefore sea spray aerosol production is zero for wind speeds less than ~ 3 m s$^{-1}$ (Blanchard, 1963; Monahan, 1971). The implication of this with respect to the findings requires discussion, for example, what is the contribution of the total CCN variability simulated between 0-3 m s$^{-1}$? At what wind speeds does the CCN concentration begin to increase sharply, is there a threshold value?

*There is no threshold velocity in the sea-spray source function, but, since the Gong parameterization is based on Monahan et al. (1986), it applies the emissions flux to be proportional to the 10m wind speed to the power 3.41, so the increase is quite steep as wind speed increases. Similarly, at the low windspeeds range mentioned (0-3 m/s) emissions fluxes will be low, and the non-linear dependence of the sea-spray emissions is one reason why this additional CCN variability becomes higher at this higher spatial resolution. We point out in the Introduction (page 3 lines 6-8) and feel this is explored sufficiently with the current text.*

• Discussion is required on the applicability of the chosen parameterisation of the resolution of the model (1Km) and time-step. Typically sea salt emission parameterisations are applicable to certain footprints, and parameterisations developed from in-situ observations are dependent on the memory of the wave field (a rising sea will result in a different emission profile than a falling sea). In addition parameterisations are developed using longer time windows for averaging for flux measurements compared to the model time-step employed. Is the sea spray source function being applied in the model at this temporal/spatial resolution in the way it was designed?

*As we state in the manuscript (page 5, lines 4-5), emissions are calculated (and enacted) every timestep of the simulation, which is 30 seconds at this high spatial resolution. Although the reviewer is absolutely correct to point out that wave state of the sea surface affects emitted sea-spray (e.g. Grythe et al., 2014, ACP), in our simulations with the Gong (2003) parameterization, these affects are not included. To address the reviewer comments we added the following to the revised manuscript:*
*"Other influences such as changes in sea surface wave state will also influence sea spray emissions (e.g. Grythe et al., 2014), but these effects are not resolved in this study. The Gong-Monahan parameterization used here is based on sea spray flux measurements made over a longer time period than the model timestep (30s), and observating capabilities now include eddy covariance sea-spray flux measurements (e.g. Norris et al., 2012), we expect our approach will resolve the dominant sources of sea spray emissions flux variability."*

**Minor comments:**

• Section 2: A figure of the modified sea-spray source function used in the study would be beneficial here, especially for experimentalists.

*We feel it is sufficient to reference the Gong (2003) paper. It's an established parameterization and was recommended for models to use in the AeroCom phase 1 co-ordinated experiment.*

• Section 2: For a modelling framework described as ground-breaking the model description is relatively sparse, for instance, how is hygroscopic growth parameterised in the model? This

will affect the evolution of the aerosol field across the domain. Please provide a more detailed description of the aerosol microphysics scheme.

*The ground-breaking aspect of this study is the ability to use a new numerical framework that is based on a coupling between the UKCA detailed aerosol module and the MetUM model at very high scale.*

*Regarding the hygroscopic growth, this aerosol process is parameterized thanks to the ZSR method (Zadanovksii, 1948; Stokes and Robinson, 1966) using data from Jacobson et al. (1996) to calculate the binary electrolyte molalities. The complete description of the hygroscopic growth parameterization as well as the description of all the others aerosol processes of the model are described in details in Mann et al. (2010).*

*We modified the manuscript in order to inform the reader that a complete description of the different aerosol processes is provided in this paper.*

• Section 2.2: It is widely known 1-moment cloud microphysics schemes introduce errors compared to 2 or3 moment schemes. Some justification of this choice is required, was it due to computational restraints?

*Yes, in this study a single-moment microphysics scheme is used as it is the only one currently available in the latest version of the Unified Model at that time. We clarified the description of the microphysics scheme in the manuscript.*

• Section 3.2, line 8: "Aitken mode are almost exclusively secondary in nature": Please reword, this is two strong, studies exist which show emission of sea spray in this size regime, e.g. Salter et al., 2015.

*We clarified the manuscript.*

• Section 4, line 25. "comprising two elements": reword.

*We clarified the manuscript.*

• Fig.4: Why do the error bars in DMS & SO2/H2SO4 not correspond? Some discussion on expected oxidation timescales required, why is there no offset between H2SO4 & DMS observed?

*We explain in the text that the steps involved for the SO2 and H2SO4 to be produced following oxidation in the atmosphere and we do not understand the reviewer's point here. We feel the existing text is sufficient here to explain what is shown in the Figure.*

• Recent studies have probed the dependence of aerosol processes on model resolution, for instance Weigum et al., 2016. This should be referenced.

*We added this reference in the introduction section.*

[revised manuscript text omitted]